# The American Monsoon System in HadGEM3 and UKESM1

Jorge L. García-Franco[1], Lesley J. Gray[1,2], and Scott Osprey[1,2]

[1]Atmospheric, Oceanic and Planetary Physics, Department of Physics. University of Oxford.
[2]National Centre for Atmospheric Science, UK.

**Correspondence:** Jorge L García-Franco: jorge.garcia-franco@physics.ox.ac.uk

**Abstract.** The simulated climate of the American Monsoon System (AMS) in the U.K. models HadGEM3 GC3.1 (GC3) and the Earth System model UKESM1 is assessed and compared to observations and reanalysis. We evaluate the pre-industrial control, AMIP and historical experiments of UKESM1 and two configurations of GC3: a low (1.875°x1.25°) and a medium (0.83°x0.56°) resolution. The simulations show a good representation of the seasonal cycle of temperature in monsoon regions, although the historical experiments overestimate the observed summer temperature in the Amazon, Mexico and Central America by more than 1.5 K. The seasonal cycle of rainfall and general characteristics of the North American Monsoon are well represented by all the simulations, showing a noticeable improvement from previous versions of the HadGEM model. The models reasonably simulate the bimodal regime of precipitation in southern Mexico, Central America and the Caribbean known as the midsummer drought, although with a stronger than observed difference between the two peaks of precipitation and the dry period. Austral summer biases in the modelled Atlantic Intertropical Convergence Zone (ITCZ), cloud cover and regional temperature patterns are significant and influence the simulated regional rainfall in the South American Monsoon. These biases lead to an overestimation of precipitation in southeastern Brazil and an underestimation of precipitation in the Amazon. The precipitation biases over the Amazon and southeastern Brazil are removed in the AMIP simulations, highlighting that the Atlantic SSTs are key for representing precipitation in the South American Monsoon. El Niño Southern Oscillation (ENSO) teleconnections, of precipitation and temperature, to the AMS are well represented by the simulations. The precipitation responses to the positive and negative phase of ENSO in subtropical America are linear in both pre-industrial and historical experiments. Overall, the biases in UKESM1 and the low resolution configuration of GC3 are very similar for precipitation, ITCZ and Walker circulation, i.e., the inclusion of Earth System processes appears to make no significant difference for the representation of the AMS rainfall. In contrast, the medium resolution HadGEM3 N216 simulation outperforms the low-resolution simulations due to improved SSTs and circulation. biases in the dynamical core, shared with HadGEM3 GC3.1 dominate.

## 1 Introduction

The American Monsoon System (AMS) is the regional monsoon associated with summer rainfall in subtropical North and South America. The AMS is associated with the coupled rainfall and circulation response to the seasonal migration of the

Intertropical Convergence Zone (ITCZ) (Zhou et al., 2016) , and is is typically subdivided into the North and South American Monsoon Systems (Vera et al., 2006). The North American Monsoon is the northernmost part of the AMS and the main source of rainfall in south-western North America, extending from central-west Mexico into the southwestern United States , with the core region located in northwestern Mexico (Adams and Comrie, 1997; Stensrud et al., 1997; Vera et al., 2006). The seasonal cycle is characterised by a wet July-August-September season and significantly drier conditions during the rest of the year

(Adams and Comrie, 1997). Several features of the North American Monsoon are modulated by the East Pacific Ocean or the Gulf of Mexico, e.g., the frequency of Gulf Surges (Douglas et al., 1993; Adams and Comrie, 1997; Seastrand et al., 2015; Lahmers et al., 2016). Moisture in the North American Monsoon is mainly advected in the low-level flow from the Gulf of California and the East Pacific Ocean whereas moisture mixed in the mid-troposphere from the Caribbean Sea and Gulf of Mexico is a secondary, but relevant, source (e.g Stensrud et al., 1997; Pascale and Bordoni, 2016; Ordoñez et al., 2019).

The South American Monsoon is a primary source of precipitation for South America, especially in the Amazon region (Gan et al., 2004; Vera et al., 2006; Jones and Carvalho, 2013). During austral summer (DJF) monsoon rainfall accounts for over 60% of the total annual precipitation in the Amazon (Gan et al., 2004; Marengo et al., 2012), whereas austral winter rainfall accounts for less than 5% of the total annual rainfall (Vera et al., 2006). The spatial domain of the South American Monsoon generally includes central and southeastern Brazil, Bolivia, northern Argentina and Paraguay but this definition can

vary amongst studies (e.g. Jones and Carvalho, 2002; Bombardi and Carvalho, 2011; Marengo et al., 2012; Yin et al., 2013).

In the central Amazon, convective activity is observed from early October but the main rainy season extends from December to April (Machado et al., 2004; Adams et al., 2013), whereas convection in southeastern Brazil starts in November and peaks in January and February (Marengo et al., 2001; Nieto-Ferreira and Rickenbach, 2011). The mean-state and variability of the Atlantic, in particular the SSTs and the Intertropical Convergence Zone (ITCZ), greatly influences the South American

Monsoon, as demonstrated in observations and climate models (see e.g. Giannini et al., 2004; Vera and Silvestri, 2009; Lee et al., 2011).

A bimodal regime characterises the seasonal cycle of precipitation in southern Mexico, Central America and the Caribbean, most commonly known as Midsummer Drought (MSD) (Magaña et al., 1999; Gamble et al., 2008), but also as "Veranillo" in Central America and "canícula" in southern Mexico (Dilley, 1996; Amador et al., 2016; Durán-Quesada et al., 2017). The

seasonal cycle in these regions is characterised by two precipitation maxima, in June and September, that are separated by a drier period in July and August. The complex interplay of SSTs, evaporation and moisture transport between the East Pacific Ocean and the Caribbean Sea are key for the spatial and temporal characteristics of the MSD (Amador et al., 2006; Herrera et al., 2015; Durán-Quesada et al., 2017; Straffon et al., 2019). Although the regions with an MSD are not formally part of the North American Monsoon, several studies (e.g Vera et al., 2006; Wang et al., 2017; Pascale et al., 2019) have analysed aspects

of the MSD in climate models and observations. This study uses the definitions for the North and South American Monsoons as previous studies (Vera et al., 2006; Marengo et al., 2012), with additional analysis on the MSD of southern Mexico and Central America (Magaña et al., 1999; Perdigón-Morales et al., 2018).

General Circulation Models (GCMs) are used to increase our understanding of monsoon dynamics and the current and future effect of greenhouse forcing on regional rainfall (see e.g. Arritt et al., 2000; Seager and Vecchi, 2010; Sheffield et al., 2013a;

Ryu and Hayhoe, 2014; Colorado-Ruiz et al., 2018). In the AMS, studies have assesed how horizontal resolution modifies the simulated climate (Pascale et al., 2016) and how climatological model biases affect simulated teleconnections (Vera and Silvestri, 2009; Bayr et al., 2019). The CMIP5 simulations of the North American Monsoon misrepresented aspects of the seasonal cycle of precipitation and overestimated the peak monsoon rainfall (Geil et al., 2013; Sheffield et al., 2013a). Most CMIP5 models simulated an earlier onset date, but improved from CMIP3 since the onset date showed a clear separation of rainy and dry seasons in daily precipitation time-series. In contrast, the simulated retreat date was unclear in most models which highlighted problems for these models to simulate the regional changes during retreat stage (Geil et al., 2013; Sheffield et al., 2013a).

The majority of CMIP5 models were unable to represent the seasonal cycle of the MSD and the total annual rainfall in Central America and the Caribbean; most models did not show signs of two-peak bimodal distribution of precipitation (Ryu and Hayhoe, 2014; Colorado-Ruiz et al., 2018). However, some models such as HadGEM2 reasonably simulated the observed bimodal regime by showing a two-peak distribution of precipitation (Ryu and Hayhoe, 2014).

The accurate simulation of the geographic distribution and seasonality of rainfall in the Amazon rainforest is a relevant issue due to the impact of the rainforest on climate and society (e.g. Li et al., 2006; Malhi et al., 2009; Yin et al., 2013). In the South American Monsoon, CMIP5 models improved from CMIP3 in the simulated distribution of precipitation during monsoon maturity and exhibited an improved seasonal cycle (Jones and Carvalho, 2013; Yin et al., 2013). However, long-term biases in the South American Monsoon, e.g., the underestimation of rainfall in the central Amazon, persisted in CMIP5 (Yin et al., 2013). The geographic distribution of rainfall during austral fall and several characteristicis of the South Atlantic Convergence Zone were also poorly represented in CMIP5. Projections from CMIP5 consistently showed a longer wet season in the South American Monsoon with earlier onsets and later retreats (Jones and Carvalho, 2013).

Climate research in recent decades has aimed to reduce uncertainty in climate projections by improving GCMs, but different approaches taken by modelling centres are seemingly disconnected (Jakob, 2014). One approach is to reduce horizontal resolution down to km resolution to rely less on parametrizations and more on physical laws to represent clouds and convection (Palmer and Stevens, 2019). A second approach aims to model Earth System processes to better characterise complex land-atmosphere-ocean biogeochemical cycles that may provide a better constraint on climate sensitivity, a parameter that depends on the carbon cycle (Marotzke et al., 2017; Sellar et al., 2019; Andrews et al., 2019). Finally, recent arguments have also suggested to include stochastic parametrisations of sub-grid processes since this approach has improved seasonal forecasts and may therefore improve climate projections (Palmer, 2019). The new phase of the CMIP project will include a range of new submissions which will include models with higher resolution and more Earth System models (Eyring et al., 2016). A comparison and evaluation of simulations with increased horizontal resolution and Earth System models may suggest where modelling efforts are resulting in significant improvements in model representation of monsoons.

The main purpose of this study is to validate the U.K. models: UKESM1, an Earth System model, and HadGEM3, the latest generation of the Hadley Centre Global Environment model. In particular, we document the main biases in these models in the region of the AMS, comparing the effect of increased horizontal resolution and Earth System processes on the representation of the AMS. The analysis may provide a framework for using these climate models in scenario studies or to further understand

**Table 1.** Summary of the datasets used in this study. For each dataset, the acronym used hereafter, the period of coverage, the field used and the horizontal resolution are shown. Some datasets extend further back in time, but only the satellite-era period is used in most of the datasets. The variables used are: precipitation, surface-air temperature ($2mT$), sea-level pressure (SLP), SSTs, the x and y components of the wind ($u$, $v$), the lagrangian tendency of air pressure ($\omega$), outgoing longwave radiation (OLR) and specific humidity ($q$).

| Dataset/ Version | Acronym | Variable | Period | Data type | Resolution | Reference |
|---|---|---|---|---|---|---|
| Global Precipitation Climatology Project v2.3 | GPCP | Precipitation | (1979-2018) | Surface station and satellite | 2.5°x2.5° | (Adler et al., 2003) |
| Global Precipitation Climatology Centre | GPCC | Precipitation | (1940-2013) | Surface station | 0.5°x0.5° | (Becker et al., 2011) |
| Climatic Research Unit TS v4. | CRU4 | Surface temperature | (1979-2017) | Surface station | 0.5°x0.5° | (Harris et al., 2014) |
| Climate Hazards Infrared Precipitation with Stations | CHIRPS | Precipitation | (1981-2018) | Surface station and satellite | 0.05°x0.05° | (Funk et al., 2015) |
| Tropical Rainfall Measurement Mission 3B42 V7 | TRMM | Precipitation | (1999-2018) | Surface station and satellite | 0.25°x0.25° | (Huffman et al., 2010) |
| Hadley Centre SST3 | HadSST | SST | (1940-2018) | Buoy and satellite | 2.5°x2.5° | (Kennedy et al., 2011) |
| European Centre for Medium-Range Forecasting ERA-5 | ERA-5 | $2mT$, SLP, $u$, $v$, $\omega$, OLR, $q$ | (1979-2018) | Reanalysis | 0.75x0.75° | (C3S, 2017) |

variability and teleconnections in this region. The remainder of this paper is organised as follows: section 2 describes the observations, reanalyses and models used, section 3 compares modelled and observed climatological features such as the ITCZ. Section 4 analyses the spatial and temporal characteristics of rainfall and convection in the AMS while section 5 documents the simulated teleconnections of ENSO. Section 6 provides a summary and discussion.

## 2 Data and methods

### 2.1 Observations and reanalysis data

Table 1 summarises relevant information of the observations and reanalysis datasets used in this study. In short, surface and satellite observations were used where available, whereas other metrics were taken from reanalysis data from the European Centre for Medium-Range Weather Forecasts (ECMWF): ERA-5, downloaded from https://climate.copernicus.eu/climate-reanalysis. Four different precipitation datasets are used.

The TRMM dataset has a high horizontal and temporal resolution and was used in several CMIP assessments (Geil et al., 2013; Jones and Carvalho, 2013) as a reliable source of precipitation (Carvalho et al., 2012). Therefore, we use TRMM as our best estimate for the spatial and temporal characteristics of the AMS rainfall. However, the period covered by TRMM

(1998-2018) is too short to analyse statistically robust teleconnections or variability, so we use GPCP, GPCC and CHIRPS for their longer period. Although a thorough validation and comparison of these datasets across the AMS domain is missing, several studies have analysed one or more of these datasets in regions of the AMS (e.g. Franchito et al., 2009; Dinku et al., 2010; Trejo et al., 2016).

## 2.2 Model data

The MOHC has submitted the output of two models for CMIP6: HadGEM3 GC3.1 (hereafter GC3) is the latest version of the Global Coupled (GC) Met Office Unified Model (UM) and UKESM1, the new U.K. Earth System Model. The most substantial change from the version used in CMIP5 (HadGEM2-AO) is the inclusion of the new GC configuration 3.1 (Walters et al., 2019) with the updated components: Global Atmosphere 7.0 (GA7.0), Global Land 7.0 (GL7.0), Global Ocean 6.0 (GO6.0), and Global Sea Ice 8.0 (GSI8.0). The GC3.1 configuration runs with 85 atmospheric levels, 4 soil levels and 75 ocean levels; for details see Williams et al. (2018) and Kuhlbrodt et al. (2018).

The GC3 model was run for CMIP6 deck experiments with two horizontal resolutions: a low resolution configuration, labelled as N96, with an atmospheric resolution of 1.875°x1.25° and a 1° resolution in the ocean model and a medium resolution configuration, labelled N216, with atmospheric resolutions of 0.83°x 0.56° and a 0.25° oceanic resolution (Menary et al., 2018).

The UKESM1 was recently developed aiming to improve the UM climate model adding processes of the Earth System (Sellar et al., 2019). These additional components include ocean biogeochemistry with coupled chemical cycles, tropospheric-stratospheric interactive chemistry which aim to better characterise aerosol-cloud and aerosol-radiation interactions (Mulcahy et al., 2018; Sellar et al., 2019). The physical atmosphere-land-ocean-sea-ice core of the HadGEM3 GC3.1 underpins the UKESM1, so that the UKESM1 and the HadGEM3 have the same dynamical core but the UKESM1 has the additional components mentioned above.

This study uses three CMIP6 deck experiments. First, the pre-industrial control (piControl) simulations, which are run with constant forcing using the best estimate for pre-industrial (1850) forcing of aerosols and greenhouse gas levels. The historical experiments are 164-yr integrations for 1850-2014 that include historical forcings of aerosol, greenhouse gas, volcanic and solar signals since 1850 (Eyring et al., 2016; Andrews et al., 2019). For further details, Andrews et al. (2020) extensively describes the historical simulations of HadGEM3-GC3.1.

In contrast to the pre-industrial control experiments, the historical experiments use time-varying aerosol and greenhouse gas emissions and land-use change (Eyring et al., 2016). In Latin-America, land-use change for agricultural purposes has dramatically decreased tree cover in Central America and south-eastern Brazil since the 1950s (Lawrence et al., 2012), thereby affecting the surface energy balance. The regional emissions of carbonaceous aerosols, nitrogen oxides and volatile organic compound in Latin America are also considered in the historical experiments. These emissions are noteworthy, e.g., due to the impact of black carbon emissions by increased biomass burning in the Amazon and northern Central America (Chuvieco et al., 2008).

The historical experiments of HadGEM3 and UKESM1 are composed of 4 and 9 ensemble members, respectively, but the results will be presented as the ensemble mean for the 1979-2014 period. These experiments will be referred to as GC3-hist and UKESM1-hist hereafter. Finally, we use the five ensemble members of the AMIP experiment from GC3 N96 covering 1979-2014. Table S1 summarises the main features of the experiments used in this study.

## 3   Climatological features

This section evaluates the simulated climatological temperature and low-level wind structure in the AMS region, as well as several characteristics of the ITCZ.

### 3.1   Temperature and low-level winds

The climatological representation of the near-surface air temperature and low-level winds in the models is compared to ERA5 in Figures 1 and 2. First, the climatology of DJF and JJA of ERA5 is shown in Figure 1a, b. The biases of the historical experiments, computed as the differences between the model and observed fields, are shown in Figures 1c, d) for GC3-hist and e, f) for UKESM1-hist. Only statistically significant differences are shown, according to a Welch t-test (Wilks, 2011), which accounts for the difference in sample size and variance between model and observations/reanalysis data. The significance for simulations with multiple ensemble members is estimated first for each ensemble member and then combined into a single probability or p-value using Fisher's method (Fisher, 1992). Pattern correlations and root-mean square error (RMSE) are shown in Figures 1c-f and in Table S2 for all seasons and more variables.

During DJF, the simulations show a colder-than-observed sub-tropical North America and a warm bias over the Amazon ($\approx 3.5$ K). The west coast of South America also shows a significant warm bias ($> 4$ K). The simulated circulation in austral summer in South America has a significant bias in the easterly flow coming from the equatorial and subtropical Atlantic. The biases in the low-level winds suggest a weaker easterly flow into southeastern Brazil but also a strong southward flow from northern to southern South America. The South America Low-Level Jet, the low-level northwesterly flow in Bolivia, observed in Figure 1a, is stronger in the simulations. This stronger than observed jet is suggestive of a stronger moisture transport to the La Plata Basin, with has been associated with a drying of the Amazon and positive precipitation anomalies at the exit region of the jet (Marengo et al., 2012; Jones and Carvalho, 2018). In turn in boreal summer (Figures 1d, f), positive biases are observed in southwestern North America ($> 3.5$ K), which are higher in UKESM1-hist than in GC3-hist. The easterly flow west of Central America has a negative bias in UKESM1 suggesting a biased flow that crosses from the Caribbean Sea into the East Pacific Ocean. Also in JJA, the simulated East Pacific surface temperatures are colder than observed for both historical experiments.

Figures 2a-d compare the GC3 piControl simulations with ERA5. In DJF, the piControl simulations show a smaller positive bias in the Amazon than the historical experiments, as well as a similar bias in the circulation in South America, with the smallest biases in GC3 N216. The inclusion of Earth System processes appears to make no improvement on the low-level circulation biases. Figures 2e, f show the difference between the historical and piControl experiment of GC3, illustrating the

response to historical forcing in GC3. The temperature response in austral summer in South America is observed as 1.5 K whereas in JJA in North America temperatures were 4 K higher in the historical experiment than in the piControl. A very similar temperature pattern response to historical forcing was observed for UKESM1 (not shown) although of slightly different magnitude. The only difference in low-level winds seem to be the easterlies in the East Pacific Ocean during JJA.

The seasonal cycle of temperature in key regions of the AMS is shown in Figure 3 which provides a better comparison of the temperature field in these experiments. These regions are illustrated in Figure 1a . The temperature in the North American Monsoon region ranges from the boreal winter 12°C to a maximum in June close to 27°C. Although colder than observed in the piControl and warmer in the historical experiments throughout the year, the models accurately reproduce the seasonal cycle, which may be relevant for the simulated monsoon onset timing and strength (Turrent and Cavazos, 2009). The piControls show a colder-than-observed winter in southern Mexico and northern Central America whereas the historical experiments show a warming signal of about 1.5 K in winter and 2 K in the summer when compared to the piControls. In spite of these biases, both types of experiments follow closely the seasonal cycle in North and Central America.

However, the seasonal cycle in South America is poorly represented in these models (Figures 3 c, d). The simulations show a stronger than observed seasonal cycle, especially the historical experiments. For example, the modelled temperature difference between late austral winter and spring was ≈4 K whereas the observed varies by less than 1 K in the same period. The models show a warm bias in the Amazon region (Fig. 3 d) which peaks in austral spring (SON), during the development of the monsoon (Marengo et al., 2012). In southeastern Brazil, the seasonal cycle is reasonably well reproduced but with a significant cold bias throughout the year which maximizes during austral winter (JJA), as models (e.g. UKESM1) simulate a temperature 4 K lower than observed. In all panels of Figure 3, the historical experiments show a significant warming signal as a response to historical forcing, which is generally stronger in UKESM1 than in GC3.

## 3.2   The ITCZ

The AMS is intertwined with the seasonal migration of the East Pacific and Atlantic ITCZ and influenced by the Walker circulation through teleconnections (Zhou et al., 2016; Cai et al., 2019). Figure 4 shows the observed and modelled climatological rainfall and the ITCZ climatological position.

Three simulations are shown: the ensemble-mean UKESM1-historical, the ensemble mean GC3 AMIP and GC3 N216-pi. Other simulations are not shown as all the coupled low resolution simulations, historical and piControl, showed very similar precipitation and ITCZ characteristics.

The observed ITCZ (Figure 4a) is found, on average, at 8°N in the East Pacific and at 6°N in the Atlantic. All the simulations reasonably represent the climatological position of the East Pacific ITCZ; however, the modelled Atlantic ITCZ near the coast of Brazil is found south of the equator at 3°S in the coupled model simulations. The GC3 N216-pi ITCZ and spatial distribution of rainfall is more consistent with the climatological position of the ITCZ of the TRMM dataset than the UKESM-hist. Rainfall near the Amazon river mouth is significantly larger in the low resolution simulations than in the TRMM dataset. However, the GC3 AMIP shows the best agreement with TRMM in ITCZ position and rainfall distribution.

The seasonal cycle of the ITCZ, precipitation rates and low-level winds in both basins are shown in Figure 5, for TRMM, UKESM1-hist, the GC3 AMIP GC3 N96-pi and GC3 N216-pi. The East Pacific (EP) ITCZ in observations (Fig. 5a) migrates southwards during the first days of the year. The EP ITCZ reaches minimum precipitation and its southernmost position at 5°N around day 100 (mid-April). During boreal spring, the ITCZ migrates northward reaching a peak latitude and maximum rainfall at 10°N by day 250, or early September. The low-level winds are predominantly easterlies, which are stronger away from the ITCZ and weaker and convergent near the ITCZ position. The position and seasonal migration of the East Pacific ITCZ is reasonably well represented in the four simulations (Figs. 5c, e, g, i), but a noticeable bias is observed in the boreal winter precipitation south of the equator in the coupled model simulations. The modelled low-level wind in the coupled model structure shows significant biases near the ITCZ. These wind biases are observed as stronger wind vectors converging toward the ITCZ during boreal summer and spring and stronger wind vectors diverging away from the equator during boreal winter.

The observed Atlantic ITCZ (Figure 5b) has a similar seasonal cycle to the EP ITCZ. The Atlantic ITCZ is close to 4°N at day 1 and migrates southwards at the start of the year reaching its southernmost position at 0° at the end of March. During boreal spring, the Atlantic ITCZ migrates north, reaching 8°N at the start of boreal summer. The boreal winter position of the modelled ITCZ is displaced with respect to the observations. The simulated ITCZ cross south of the equator during boreal winter covering 10-0°S with rainfall rates above 12 mm day $^{-1}$. After boreal spring, the modelled ITCZ crosses back north of the equator and matches the observed ITCZ reasonably well for boreal summer and fall. Low-level wind biases near the Atlantic ITCZ (Figures 5f and h) show that north of the equator the models show a stronger than observed northward wind, and a stronger northerly flow south of 10°S. The biases in the Atlantic ITCZ can also be observed in the overturning circulation (Figure S1) and the associated Walker circulation as significant negative $\omega$ and $q$ biases just north and south of equatorial South America indicative of weaker convective activity. The Atlantic Ocean shows a biased strong ascent south of the equator and a biased weak ascent north of the equator in the low resolution simulations. These biases described above were found to be of similar magnitude in the coupled model simulations run at N96 resolution, both historical and piControl experiments, however, these biases improved in the medium resolution GC3 N216-pi and in the AMIP simulations (Figures 5f, j).

The South Atlantic Convergence Zone (SACZ) is a nortwest-southeast oriented band of convection and is a prominent influence on the South American Monsoon mean and extreme rainfall (Carvalho et al., 2004; Marengo et al., 2012). UKESM1 and GC3 appear to reasonably simulate the spatial pattern of active SACZ days and the seasonal cycle of SACZ activity (Figure S2).

## 4 The American Monsoon System

### 4.1 Mean seasonal precipitation

The austral summer (DJF) rainfall distribution and biases in South America are shown in Figure 6 for GC3 N216, UKESM-hist and GC3 AMIP. The maximum austral summer rainfall in TRMM (Figure 6a) is found as a nortwest-southeast oriented band of precipitation from the core Amazon region into southeastern Brazil. The coupled simulations (e.g. Figure 6b, c) overestimate rainfall in southeastern Brazil and underestimate rainfall in the core Amazon region.

The biases are illustrated (Figures 6e-h) as the precipitation difference between the simulations and TRMM . The coupled simulations show three main biases. Rainfall in the Atlantic ITCZ in these simulations is displaced southwards, observed as positive (+5 mm day$^{-1}$) biases south of the equator and negative biases (-5 mm day$^{-1}$) north of the equator in the Atlantic. Second, the models underestimate rainfall in the core Amazon basin by -3 mm day$^{-1}$ on average, whereas rainfall in south-eastern Brazil is overestimated by more than +5 mm day$^{-1}$, approximately +100% of the observed rainfall in this region. The precipitation biases are associated with a stronger northerly flow in South America, transporting moisture from the Amazon into southeastern Brazil and the La Plata Basin. The magnitude of these biases is smaller in GC3 N216 (Figure 6f) than in the low resolution simulations, such as UKESM1-hist. The ensemble mean GC3 AMIP (Figure 6d) shows a better representation of the austral summer rainfall patterns, removing the main biases (Figure 6g) of the coupled simulations. The response to historical forcing, illustrated by the difference between UKESM1-hist and UKESM1-pi (Figure 6h), is much weaker than the magnitude of the biases.

The modelled and observed JJA mean rainfall and biases for Mexico and Central America are shown in Figure 7. The main feature (Figure 7a) is the East Pacific ITCZ which extends north to 15°N near the western coast of Mexico as a broad band of rainfall (>11 mm day$^{-1}$). The North American Monsoon can be observed as a band of precipitation mainly across western Mexico. In the core monsoon region, near the Sierra Madre Occidental (Adams and Comrie, 1997; Zhou et al., 2016), the JJA-mean rainfall is higher than 6 mm day$^{-1}$.

The modelled East Pacific ITCZ (Figures 7e, f, g) rainfall is overestimated by more than 5 mm day$^{-1}$, even more so in GC3 AMIP. This wet bias is associated with an easterly bias in the low-level circulation, suggesting a weaker flow from the Caribbean into the East Pacific. The low-resolution simulations (Figure 7e) underestimate rainfall (-5 mm day$^{-1}$) over land in southern Mexico, Guatemala and Belize. Rainfall in the Caribbean islands and Florida is underestimated (-1 mm day$^{-1}$) in all simulations. The distribution of rainfall in the North American Monsoon region is relatively well represented in all the simulations, showing only a small wet bias (+2 mm day$^{-1}$) in western Mexico. The northernmost part of the North American Monsoon (southwestern US) is best simulated by GC3 N216-pi. In most cases for JJA in this region, the precipitation and wind biases were reduced in the high-resolution simulation (Figure 7f). The precipitation response to historical forcing is much lower than the biases (Figure 7h) with no significant precipitation differences over land.

## 4.2 The annual cycle of rainfall

Figure 8 shows the pentad-mean cycle of rainfall over the North American Monsoon, the Midsummer drought (MSD), the Amazon and Eastern Brazil regions. The correlation between TRMM and model and reanalysis data is also shown in each panel. The seasonal cycle of precipitation in the MSD region in the simulations is well represented as all the simulations show the characteristic bimodal distribution. However, the characteristics of the simulated MSD are different to observations. For example, the magnitude of the first peak in the simulations is higher than TRMM by 4 mm day$^{-1}$. Similarly, the differences between the first peak and the MSD and between the MSD and the second peak are more pronounced in the coupled simulations. The timing of the MSD period is different in the models, as the simulations show the driest period taking place 10 days after

TRMM and ERA5. All the simulations show different magnitudes of the first and second peak and the MSD precipitation, including the AMIP simulation that overestimates the second maximum of rainfall by 2-3 mm day$^{-1}$.

Rainfall in the North American Monsoon (Figure 8b) show a sharp increase of rainfall around mid-June in models and observations, suggesting that onset timing and strength is well represented in these models. Moreover, the modelled and the observed mean rainrates during monsoon maturity is 4 mm day$^{-1}$, from mid-July until early September. The historical simulations show a shorter wet season characerised by an earlier retreat of the rainfall and, as all the simulations, a positive boreal fall rainfall bias (+1 mm day$^{-1}$), a feature that has been shown in these models in CMIP5 (Geil et al., 2013).

The seasonal cycle of precipitation in eastern Brazil is characterised by a very wet summer ($\sim$8 mm day$^{-1}$) compared to a very dry ($\sim$0.2 mm day$^{-1}$) winter (Figure 8c). Austral summer rainfall in the observations consistently shows that maximum rainfall is found in early January ($\sim$8 mm day$^{-1}$). Rainfall in this region decreases to $\sim$6 mm day$^{-1}$ by late March as the monsoon migrates northward and sharply decreases in austral fall. The models (Figure 8c) show a positive bias during monsoon maturity. This bias was found to be of +4 mm day$^{-1}$ and +2.5 mm day$^{-1}$ for the low and medium resolution simulations, respectively. The bias in the seasonal cycle is consistent with the biases shown in Figure 6, which showed that rainfall in southeastern Brazil is overestimated, especially in the low resolution coupled simulations. In contrast to the coupled simulations, GC3 AMIP shows a very good agreement with the observed maximum summer rainfall and the seasonal cycle (r=0.978) throughout the year.

Finally, the simulated rainfall in the Amazon in the coupled simulations show a dry bias in the austral summer and a good agreement with the observations during austral winter (Figure 8d). The models also represent, with reasonable skill, the transition from early austral spring (4 mm day$^{-1}$ in September) to summertime rainfall (6 mm day$^{-1}$ in November). However, peak summertime rainfall is underestimated by the coupled model simulations, particularly the historical experiments. Rainfall in the Amazon from January to March, in both TRMM and ERA-5, is close to 10 mm day$^{-1}$, yet the low resolution simulations show rainfall rates of 8 mm day$^{-1}$ in mid-February. GC3 N216-pi shows a better agreement with observations but still underestimates summertime rainfall by 1 mm day$^{-1}$. The dry Amazon bias has been a know feature of GCMs, including the MOHC models since CMIP3 (Li et al., 2006; Yin et al., 2013). In these simulations the dry Amazon bias is only alleviated in GC3 AMIP whose seasonal cycle and maximum summer rainfall agree well with observations.

## 4.3 Characteristics of convective activity

The seasonal cycles of out-going longwave radiation (OLR), vertical velocity ($\omega$) and specific humidity ($q$) are key features of a monsoon since these quantities characterise the strength and height of deep convection. Figure 9 shows the pentad-mean annual cycle of OLR, $q$ and $\omega$ at the 500-hPa level in four regions of the AMS. For the North American Monsoon the seasonal cycle of OLR, $q$ and $\omega$ is relatively well represented in the simulations. During late boreal winter and early spring, OLR increases steadily as a result of surface warming. However, in early June, near the onset date (Douglas et al., 1993; Geil et al., 2013), OLR sharply decreases reaching a minimum value of 246 W m$^{-2}$ by mid-July. The vertical velocity decreases steadily from January to a minimum in August, indicating ascent from May 1st until September 15th. The models show similar seasonal

cycles but overestimate the summertime OLR by $\approx 6$ W m$^{-2}$ and underestimate mid-level moisture by 0.3 g/kg and $\omega$ by 0.01 Pa s$^{-1}$. The simulated shallower convection and drier mid-troposphere is seemingly compensated by stronger ascent.

In the MSD region, OLR and $q$ show signs of convective activity from mid-April, as OLR sharply decreases and moisture increases. The characteristic MSD bimodal distribution of precipitation can also be observed as two peaks of low OLR, high $q$ and low $\omega$. These periods are separated by a period of relatively higher OLR, lower $q$ and weaker ascent from June 15 until late 310 August. Arguably with a small dry bias with shallower convection after mid-July, the simulations follow closely the observed seasonal cycle. The simulated first peak of rainfall has similar OLR and mid-level moisture but stronger ascending motions, which may explain the positive rainfall bias in this period showed in Figure 8a. In the period between the first peak and the MSD, the simulated OLR increase more sharply than observations from 220 W m$^{-2}$ (June 15) to 250 W m$^{-2}$ (early August), with similar behaviour in $\omega$ and $q$. The period during the second peak of rainfall in September shows signs of shallower 315 convection and a drier mid-level when compared to ERA5.

In southeastern Brazil, the simulations reasonably follow the annual cycle of OLR, $q$ and $\omega$ of the reanalysis, particularly during austral winter. The observed $q$ in the dry seasons of austral fall, winter and spring in ERA5 is very similar to the simulated $q$. However, during austral summer, the coupled model simulations show significant biases characterised by stronger ascent and increased specific humidity in the mid-levels, although the height of convection (OLR 225 W m$^{-2}$) is only modestly 320 higher in the simulations.

The simulated OLR, q and $\omega$ exhibit the highest biases in the Amazon. During austral summer, particularly January and February, the simulated convective activity is shallower (OLR bias of +25 W m$^{-2}$) and weaker (positive $\omega$ bias +0.02 Pa s$^{-1}$) and the mid-level troposphere is drier ( -0.5 g/kg) than in ERA5. In spite of biases in the magnitude of OLR, $q$ and $\omega$ during peak convective activity, the seasonal variation is very well simulated so that convective activity, as evidenced by these metrics, 325 starts and ends in the simulations within one or two pentads of the reanalysis. The smallest biases in coupled simulations are those of GC3 N216-pi, for this and the other regions. The simulated OLR, $q$ and $\omega$ by GC3 AMIP in southeastern Brazil and the Amazon show a much better agreement with the reanalysis during austral summer than the rest of the observations.

## 5  ENSO Teleconnections

El Niño-Southern Oscillation (ENSO) teleconnections are the prominent source of interannual variability in the AMS (Vera 330 et al., 2006). This section shows the temperature, sea-level pressure (SLP) and precipitation responses to observed and simulated ENSO events in the AMS. ENSO events were defined when the DJF-mean Niño 3.4 index was above or below 0.65 (Trenberth, 1997). Other indices, including the use of a 5-month running mean (Trenberth et al., 1998), were tested without significantly changing the results. Previous studies (e.g. Menary et al., 2018; Kuhlbrodt et al., 2018) showed that the MOHC models reasonably simulate several characteristics of ENSO such as the period and SST patterns.

The temperature and SLP response to ENSO events is shown in Figure 10 for model and ERA5 data. The modelled warm anomaly during El Niño events in the East Pacific Ocean does not extend to the east as much as the observed warm anomaly and the cold anomalies during La Niña events in the Central Pacific are colder in the simulations. However, the simulated and

observed teleconnection patterns to South America, e.g., the cold anomalies during La Niña events in northern South America are seemingly well simulated. The teleconnection to southern North America, i.e., colder (warmer) conditions in southern (northern) North America during El Niño events are relatively well simulated even though the low resolution simulations showed a broader and stronger than observed response in southeastern US.

The SLP response in the northern Pacific and North America, known as the Pacific North-American pattern, is linked with a displacement of the subtropical jet affecting the eastward propagating wave activity that reaches the North Atlantic (e.g. Bayr et al., 2019; Jiménez-Esteve and Domeisen, 2020). During ENSO events, the Aleutian Low is strengthened in ERA5, with a strong SLP anomaly (-4 hPa) off the coast of California. The models show a similar but smaller SLP response in the same region. Positive ENSO events are typically associated with a negative phase of the North Atlantic Oscillation (NAO), with an opposite response for La Niña events. While the models seem to be able to capture this response of the NAO, the simulated response is weaker than observed. A sensible representation of the ENSO-NAO tropospheric teleconnection may be relevant to then simulate the effect of the NAO on Central American and northern South American rainfall (Giannini et al., 2000, 2004).

The rainfall anomalies associated with ENSO events are shown in Figure 11. Three regions in the AMS have a significant precipitation response to ENSO events in the observations and simulations. In southern North America, rainfall increases (decreases) during El Niño (La Niña) events due to the effect of Rossby waves on the subtropical jet and wintertime midlatitude disturbances (Vera et al., 2006; Bayr et al., 2019). The GPCP dataset (Figure 11a, b) shows significant boreal winter rainfall increases in southeastern US and the Gulf of Mexico during El Niño events, and an opposite response to La Niña phases. All the simulations reproduce this teleconnection rainfall pattern.

The anomalies in the Amazon show the strongest response to ENSO events in the observations. This teleconnection works through the coupling of ENSO with the Walker circulation (Vera et al., 2006; Cai et al., 2019), illustrated in Figure S3. Significant positive (negative) rainfall anomalies during the negative (positive) phase of ENSO in northern South America are observed in GPCP. All the simulations show a very similar and statistically significant response. The models also simulate the observed response in southeastern South America (SESA) of positive anomalies during El Niño and negative anomalies during La Niña events.

Figure 12 shows the observed and simulated precipitation responses in four regions of the AMS to different magnitudes of ENSO events, essentially showing the degree of linearity of ENSO teleconnections to the AMS. While the observed response shows some degree of linearity for El Niño events in South America (panels c, d), the majority of the observed responses, particularly to La Niña phases, are not linear. However, the simulations show several signs of linearity; for instance the historical experiments exhibited a linear response in precipitation to ENSO events in North America and SESA. However, some simulated responses, e.g. to La Niña phases in South America in the piControl simulations, show signs of non-linearity.

The different observed SST patterns for each ENSO event are a source of non-linearity of ENSO impacts over South America (Sulca et al., 2018; Cai et al., 2020). Principal component analysis has shown that ENSO events may be separated into two categories: Central Pacific (CP) and East Pacific (EP) events (Cai et al., 2020). Figure S4 shows that both UKESM1 and GC3 reasonably simulate the observed SST patterns associated with EP and CP El Niño events. Furthermore, Figure 13 compares

the precipitation anomalies for each type of ENSO event in observations with three simulations: GC3 N96-pi, GC3 N216 and GC3 AMIP.

The observed precipitation response in the GPCC dataset to EP La Niña over equatorial South America is not significant and is smaller than the observed strong positive precipitation response to CP La Niña events in the same region. However, the simulated response in GC3 N96-pi and GC3 N216 during La Niña events appears to be independent of the type of event. In contrast, GC3 AMIP does show a positive, and significant, anomaly for CP La Niña events and weaker and not significant anomalies during EP events. The observed response to El Niño events in GPCC is also dependent on the type of event. EP EL Niño events show significant negative anomalies over the Amazon and positive anomalies over SESA whereas CP events only show significant anomalies (-1 mm day$^{-1}$) over northeastern South America. While the coupled models (GC3 N96-pi and GC3 N216) do show a stronger response to EP EL Niño events than to CP events, the patterns of the response are very similar. In contrast, GC3 AMIP shows a very strong negative response to EP El Niño events in the Amazon but the response to CP events is much weaker and is only significant in northeastern South America. In other words, GC3 AMIP agrees well with the observed non-linear teleconnection patterns whereas the teleconnections in the coupled models do not depend on the type of ENSO event.

## 6 Summary and discussion

This study analysed results from the MOHC models, HadGEM3 and UKESM1 in their pre-industrial control, historical and AMIP experiment contributions to CMIP6. In particular, we focused on evaluating the modelled climate of the AMS comparing the effect of including Earth System processes or increasing resolution for representing regional rainfall. A schematic in Figure 14 shows the primary components of the AMS climate and summarizes the main biases in these simulations.

Rainfall in the North American Monsoon was particularly well simulated by the models. The seasonal cycle, peak monsoon rainfall rates and timings of monsoon onset and retreat in the simulations agreed well with TRMM. The historical experiments overestimate the mean temperature in most of the Americas by 1.5 K, but particularly in boreal summer in southwestern North America (+4 K). In spite of this warm bias, the temperature seasonal cycle is well represented by these models. These results suggest model improvement on the simulation of the North American Monsoon from previous versions of the MOHC models (Arritt et al., 2000), and most of the model cohorts of CMIP3 and CMIP5 (Geil et al., 2013). For example, most of CMIP5 models showed a very wet bias during monsoon maturity whereas rainfall during monsoon maturity in all the experiments of this study are within 1 mm day$^{-1}$ of observations. However, these models continue to show biases during monsoon retreat as rainfall does not decrease as sharply as in observations after mid-September.

The Midsummer Drought (MSD) of southern Mexico and Central America is a regional feature of precipitation that most of CMIP5 models had difficulty capturing, except for instance for the MOHC models (Ryu and Hayhoe, 2014). The MSD in UKESM1 and GC3 continues to be relatively well represented, although with some differences in the timing and strength of the bimodal cycle. The models simulate a wetter-than-observed first peak of precipitation and a drier MSD period, therefore simulating a higher difference between the first peak and the dry period. The so-called second peak of precipitation found in

late August is simulated in close agreement with TRMM, except in the AMIP experiment. Rainfall during the first peak has been too wet in these models since CMIP3, suggesting a persistent wet bias in this region associated with the East Pacific ITCZ (Ryu and Hayhoe, 2014; Mulcahy et al., 2018).

The East Pacific ITCZ migration and position was shown to be relatively well represented by the models (Figs. 4 and 5). However, the models showed an overestimation of boreal summer rainfall near the coast of Central America (Figure 8). These biases are associated with an easterly bias in the low-level wind, suggesting a bias in the flow from the Caribbean Sea into the Eastern Pacific which is relevant for moisture transport and controlling the SSTs (Herrera et al., 2015; Durán-Quesada et al., 2017). The simulations also showed a biased Atlantic ITCZ that was displaced south of the observed ITCZ position during boreal winter (Figure 5), particularly in the low resolution coupled simulations.

In the Amazon, the simulations showed a warm bias (+2 K) during austral spring and summer, a typical feature of previous models (Jones and Carvalho, 2013), and a colder than observed southeastern Brazil. These biases were linked with decreased cloud cover and less rainfall over the Amazon and more high clouds and rainfall in southeastern Brazil (Figures 7 and 9). The low cloud cover, warm and dry Amazon biases are intertwined with the low-level circulation from the Atlantic into the South American continent. The biases in the circulation during austral summer were observed as a northerly flow anomaly over the central and southern Amazon, a feature that has been associated with a stronger moisture transport away from the Amazon (Marengo et al., 2012; Jones and Carvalho, 2018). During the period of maximum mean rainfall rates in February, the simulations can overestimate rainfall by 3 mm day$^{-1}$ in southeastern Brazil and underestimate rainfall in the Amazon by a similar rate. The historical experiments showed a small drying response to historical forcing in the Amazon therefore slightly increasing the magnitude of this dry bias. The AMIP simulation with the SST biases removed improved the Atlantic ITCZ representation and the precipitation, cloud cover and temperature biases over the South American Monsoon. The improvement in the circulation and precipitation biases in the AMIP simulation suggest that the origin of the dry Amazon bias are the biases in the Atlantic SSTs.

The canonical teleconnection responses of temperature, SLP and precipitation in the AMS to ENSO events are well represented in these models. The positive (negative) anomalies observed in northern Mexico and South Eastern South America during El Niño (La Niña) events are well represented in these experiments. Similarly, the teleconnection to the Amazon is well represented for both phases of ENSO, in spite of relevant biases in the region. ENSO teleconnections in these simulations were found to be approximately linear, i.e., the precipitation response is linearly related to the magnitude of the SST perturbation in the EN 3.4 region. In this model framework, positive and negative phases produce the opposite and equivalent precipitation response in the AMS. In contrast to observations and the GC3 AMIP simulation, the precipitation response in the coupled models appears to be independent of separating ENSO events into Central and East Pacific events. The fact that these models show a reasonable representation of ENSO diversity in SST patterns but the models do not replicate the observed non-linear dependance to ENSO events warrants further analysis.

The main biases are smaller in the medium resolution GC3 N216 compared to the low resolution experiments. In contrast, including Earth System processes in the UM model only affects the surface temperature response to historical forcing and not the dynamical biases that drive the precipitation and ITCZ biases. In short, the main dynamical biases in UKESM1 are very

similar to those in GC3 N96 as these two models share the same dynamical core and only when resolution is increased are these biases reduced significantly. In spite of not improving the dynamic representation of the AMS, UKESM1 does show a stronger temperature response to forcing, as this model has a higher climate sensitivity than GC3 (Andrews et al., 2019; Sellar et al., 2019). A relevant difference between UKESM1 and GC3 is that warming over the historical period in Mexico and the Amazon is higher in UKESM1 than in GC3. This warming may be a consequence of the land-use change in these regions playing a role in the UKESM1 representation of soil-atmosphere feedbacks.

The improvement in the medium resolution simulation may largely be due to the improved dynamics of the ocean or the atmosphere. For example, the Atlantic ITCZ biases have been shown to be directly affected by processes in the convective scheme (Bellucci et al., 2010), such as the treatment of entrainment and moisture-cloud feedbacks (Oueslati and Bellon, 2013; Li and Xie, 2014). The resolution of the ocean model has been shown to impact the eddy heat flux parametrisation and the associated heat uptake and transport of the ocean (Kuhlbrodt et al., 2018). The improvement in the Atlantic SSTs and ITCZ and the associated dynamics also improves the associated circulation biases and moisture transport in the South American Monsoon. In other words, the oceanic resolution may play an important role in the cross-equatorial heat and moisture transport, SST gradients and the land-sea circulation over the Amazon during austral summer that is key for representing the geographic distribution of rainfall in South America.

These CMIP6 simulations of HadGEM3 and UKESM1 show some signs of model improvement, particularly in the North American Monsoon and may be used to better understand the response to current and future response to anthropogenic forcing. Furthermore, several aspects of the climate of the AMS that are well simulated by these models, such as a good representation of the MSD and a resonable representation of ENSO diversity, may suggest further use of these simulations to address outstanding questions of climate variability in this region across different temporal scales.

*Author contributions.* JLGF conducted the analyses, LJG and SO directed the research. All authors were fully involved in the revisions and the preparation of the paper.

*Competing interests.* The authors declare that there are no competing interests.

*Data availability.* ERA5 data was made available by Copernicus at https://cds.climate.copernicus.eu whereas the model data is available in the CMIP6 Earth System Grid Federation (ESGF) at https://esgf-index1.ceda.ac.uk/projects/cmip6-ceda/.

*Acknowledgements.* JLGF was funded by an Oxford-Richards Scholarship.The authors thank David Adams and one anonymous reviewer for their very helpful comments that have improved this manuscript. The authors would also like to thank Robin Chadwick, Tim Woollings, Antje Weisheimer and Gabriel Martins Palma Perez for their very helpful suggestions and useful discussions.

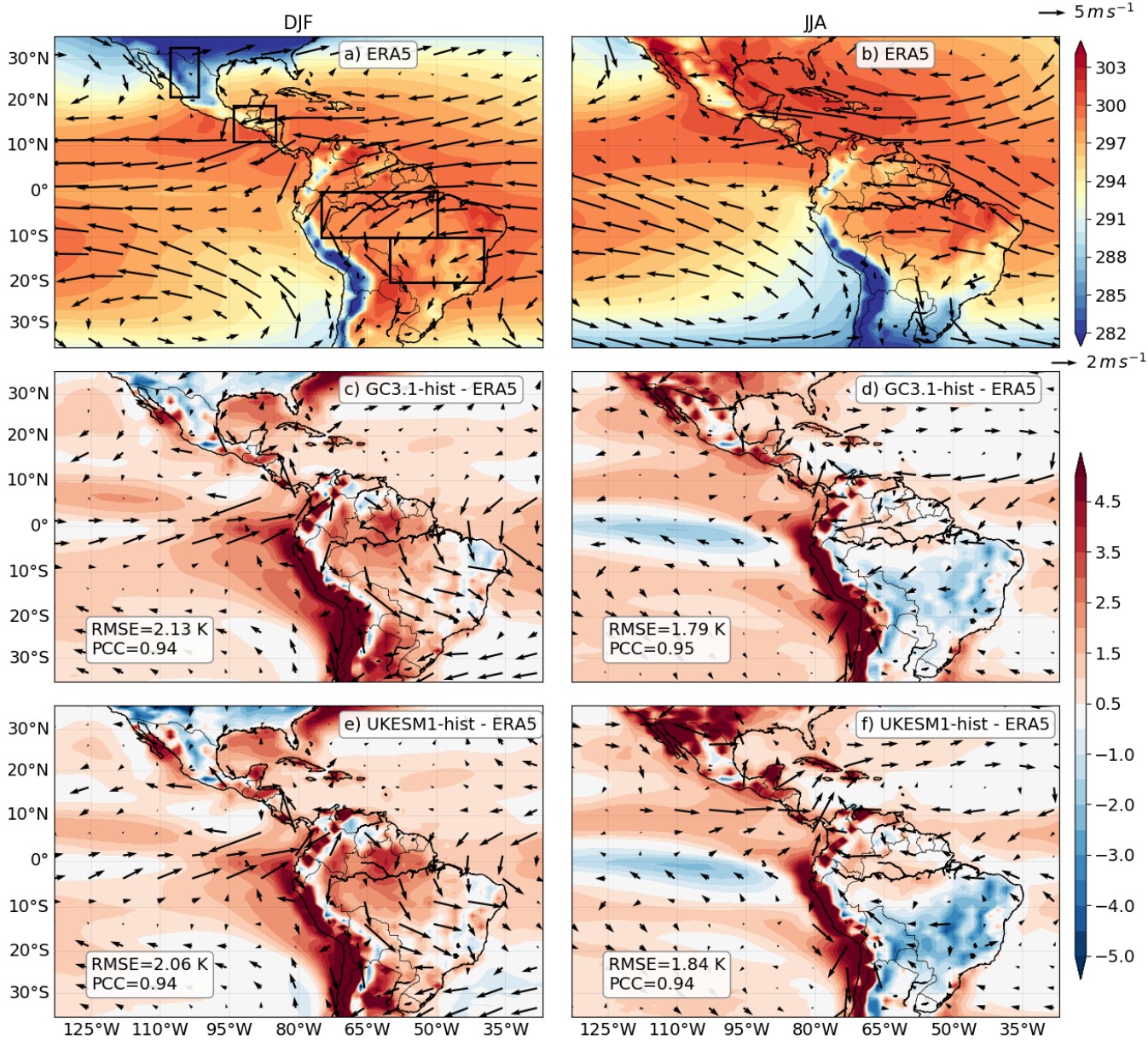

**Figure 1.** (a, b) Temperature (color-contours in K) and wind speed (vectors) at 850 hPa DJF and JJA climatogies in ERA5. The biases are shown as the differences between the ensemble mean from the historical experiment of (c, d) GC3 and (e, f) UKESM1 and ERA5. The climatogies and biases are shown for (a, c, e) boreal winter (DJF) and (b, d, f) boreal summer (JJA). Only differences statistically significant to the 95% level are shown, according to a Welch t-test for each field. The key for the size of the wind vectors is shown in the top right corner of panels b) and d). The root-mean square error (RMSE) and pattern correlation coefficient (PCC) are shown on the bottom left of c-f.

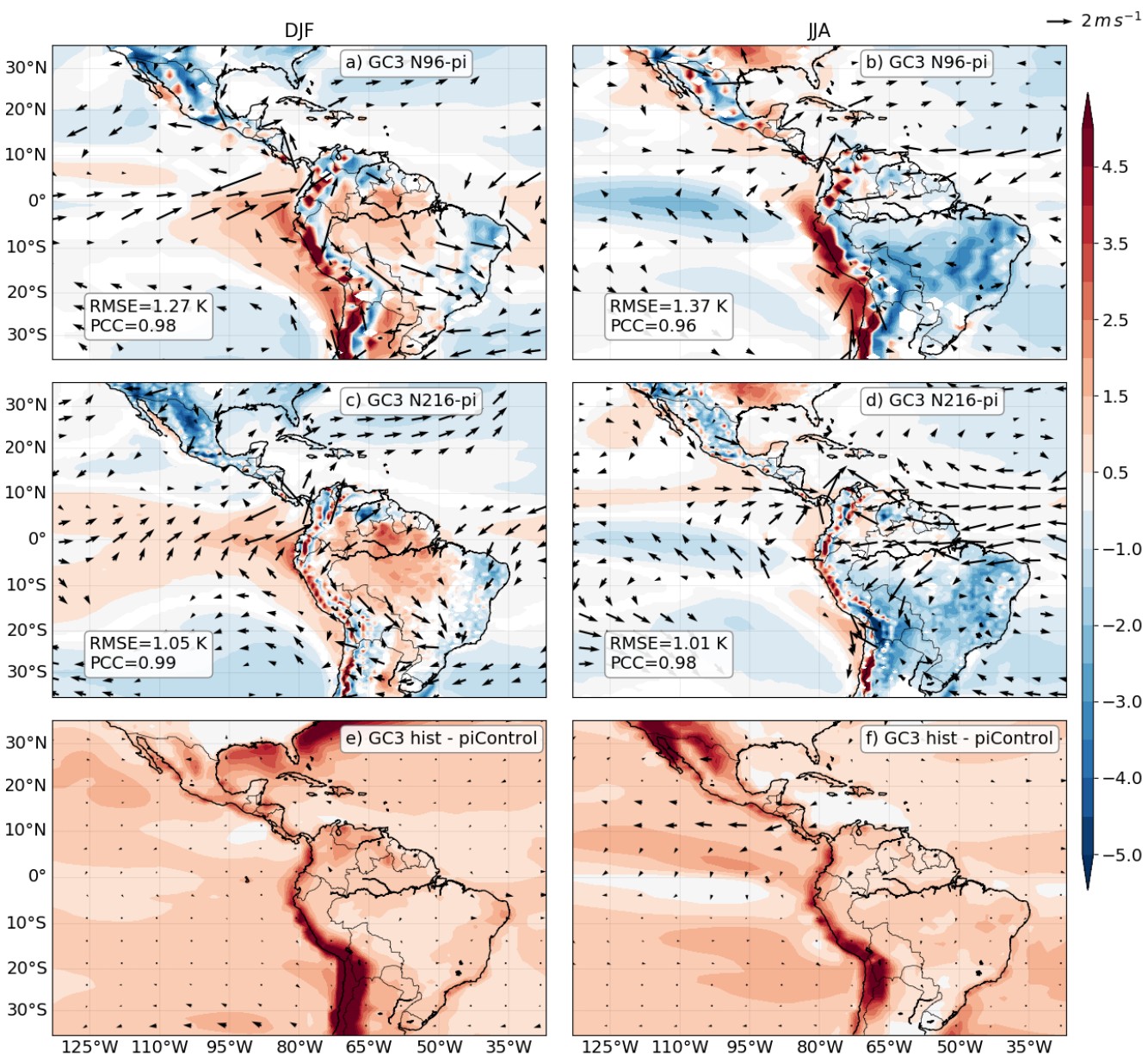

**Figure 2.** As in Figure 1, but showing the differences between the piControl simulations of (a, b) GC3 N96-pi and (c, d) GC3 N216-pi, and ERA5. (e, f) show the statistically significant differences between the historical (1979-2014) and piControl experiments of GC3. The RMSE and PCC are shown on the bottom left of a-d.

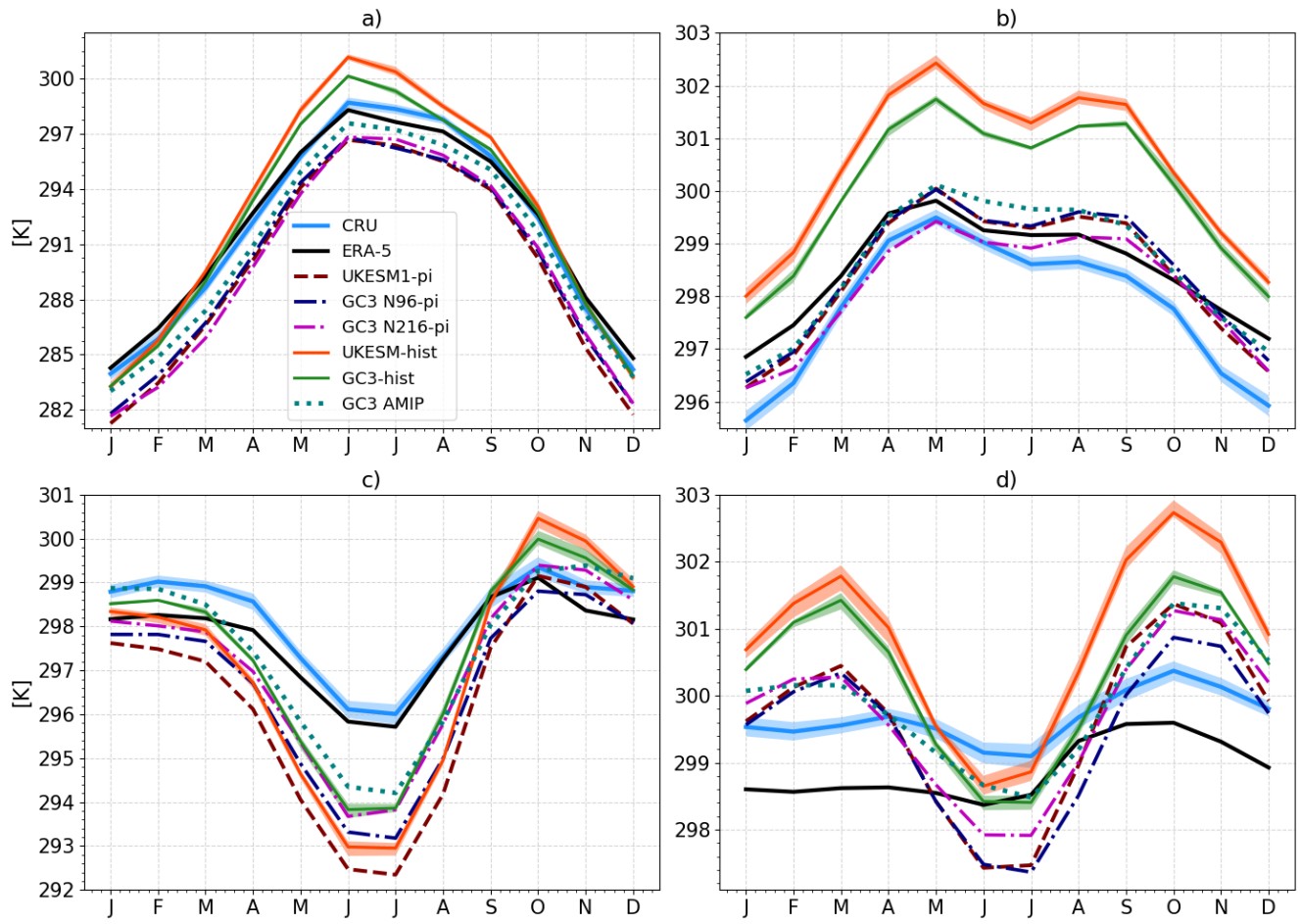

**Figure 3.** Monthly-mean temperature in the (a) North American Monsoon [19-35°N,110-103°W], (b) the Midsummer drought [11-19°N,95-85°W] (c) Eastern Brazil [20-10°S,60-40°W] and (d) the Amazon basin [-10-0°S,75-50°W] regions. The shadings for the CRU dataset represents the observational uncertainties and for the historical simulations the shading is the ensemble spread. The regions for this plot are shown in Figure 1a.

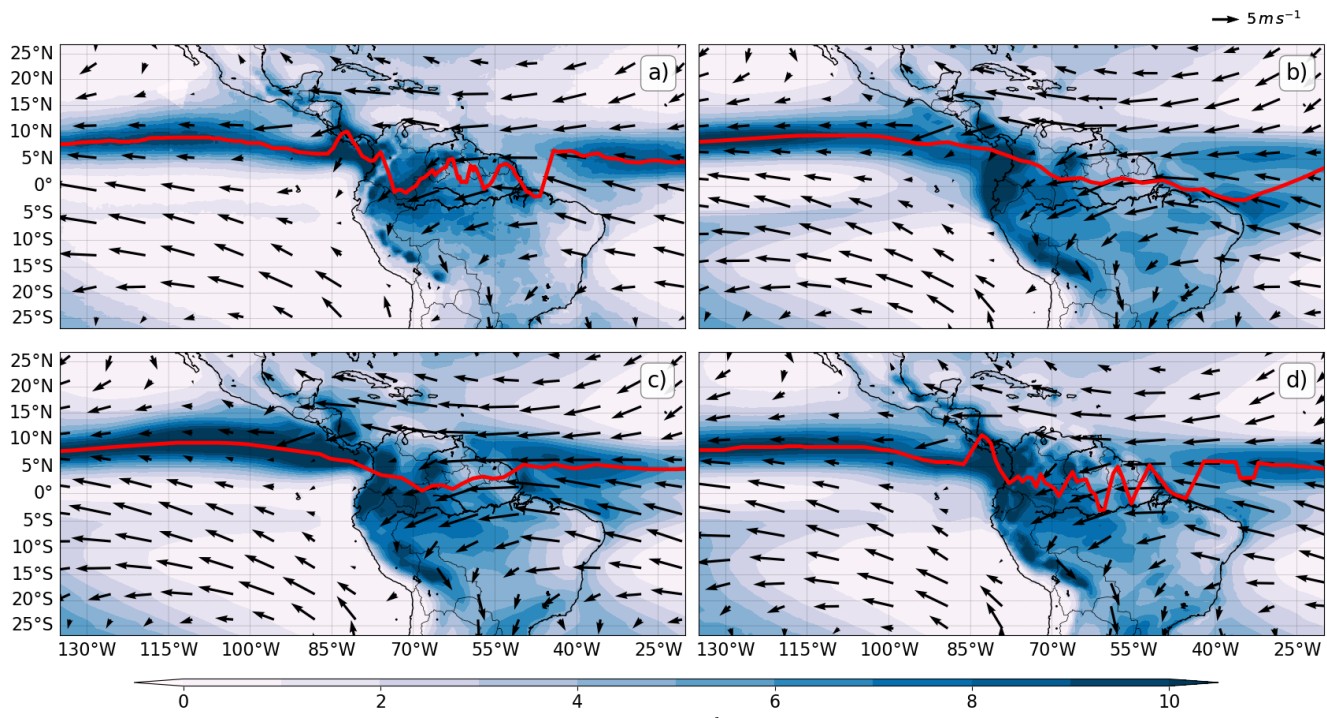

**Figure 4.** Climatological rainfall [mm day$^{-1}$] and low-level wind speed (850-hPa) in (a) TRMM and ERA-5, (b) the ensemble-mean UKESM-historical, (c) GC3 AMIP and (d) GC3 N216-pi. The red line highlights the maximum rainfall for each longitude as a proxy for the position of the ITCZ.

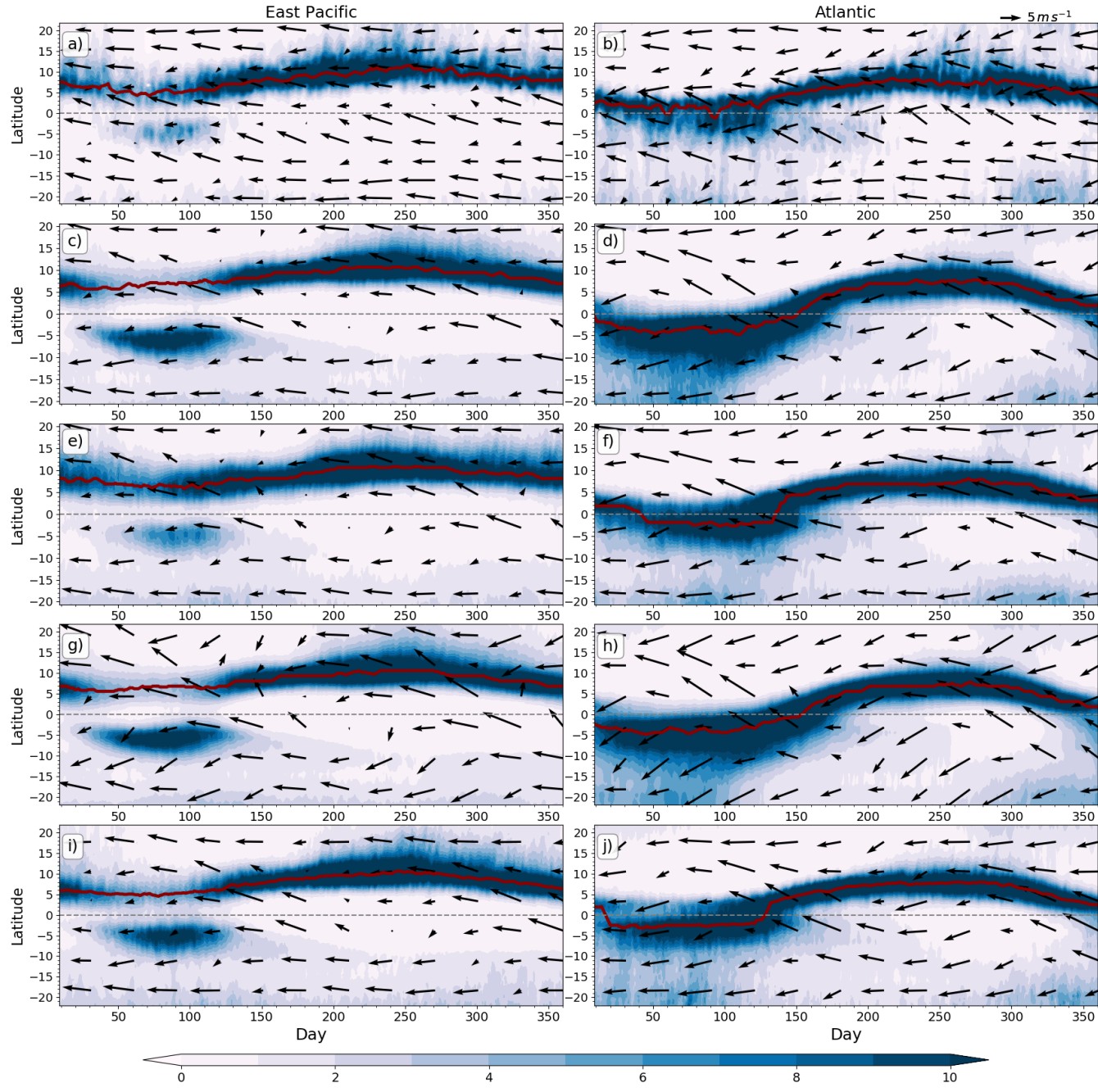

**Figure 5.** Time-Latitude plot of daily mean rainfall (colour contours) and low-level wind speed (850 hPa) longitudinally averaged over the (a, c, e, g) East Pacific [150°W-100°W] and (b, d, f, h) Atlantic [40°W-20°W] Oceans. (a, b) show rainfall from TRMM and winds from ERA-5, (c, d) the ensemble-mean UKESM-historical, (e, f) GC3 AMIP, (g, h) N96-pi and (i, j) GC3 N216-pi. The red solid line shows the ITCZ as the latitude of maximum precipitation.

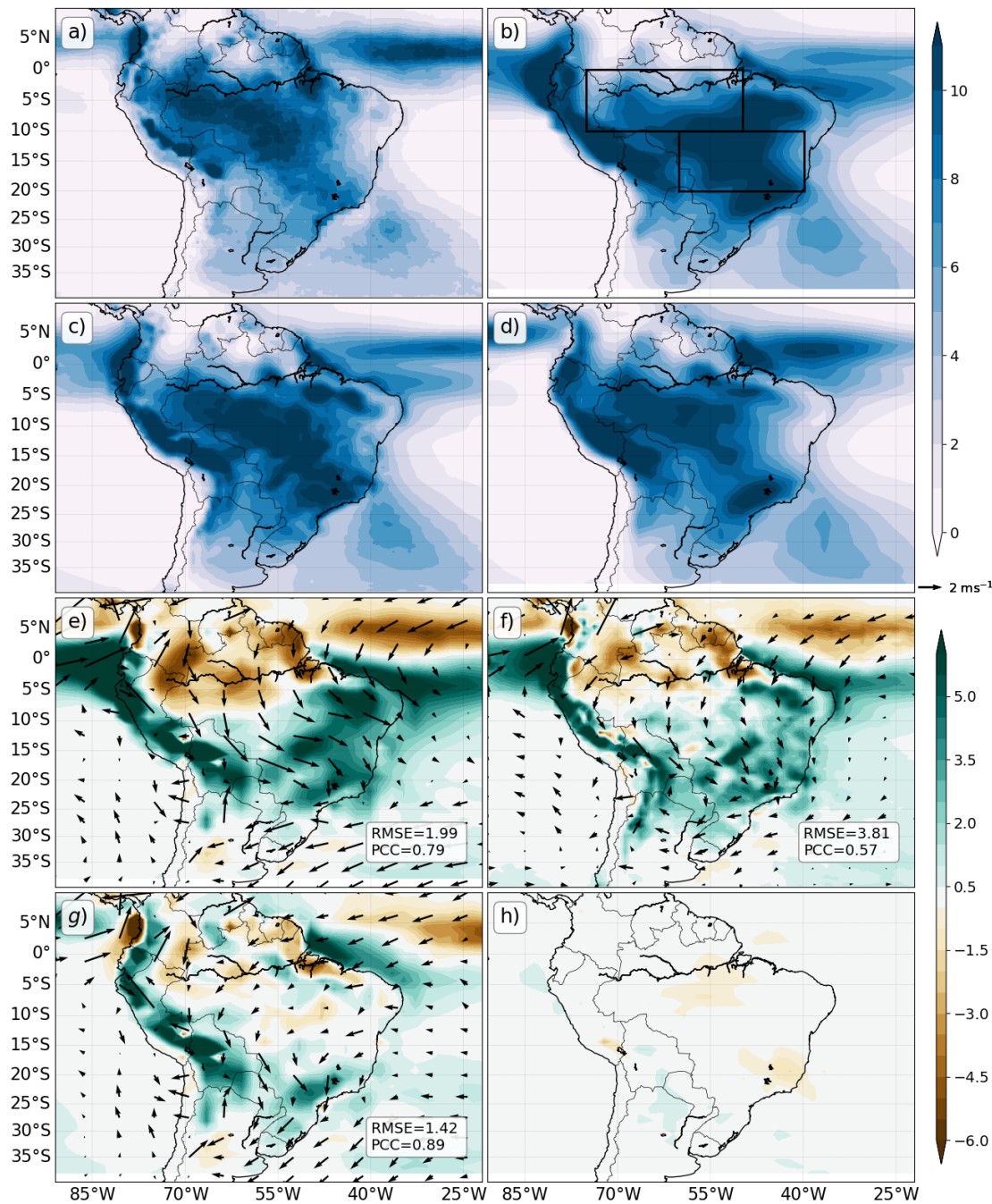

**Figure 6.** DJF mean rainfall [mm day$^{-1}$] from (a) TRMM, (b) UKESM1-historical, (c) GC3 n216 and (d) GC3 AMIP. (e, f, g) show the statistically significant differences between panels (b, c ,d) and (a) TRMM, respectively. (h) Precipitation difference between UKESM-historical and UKESM1-pi, only statistically significant differences (95%) confidence level is shown.

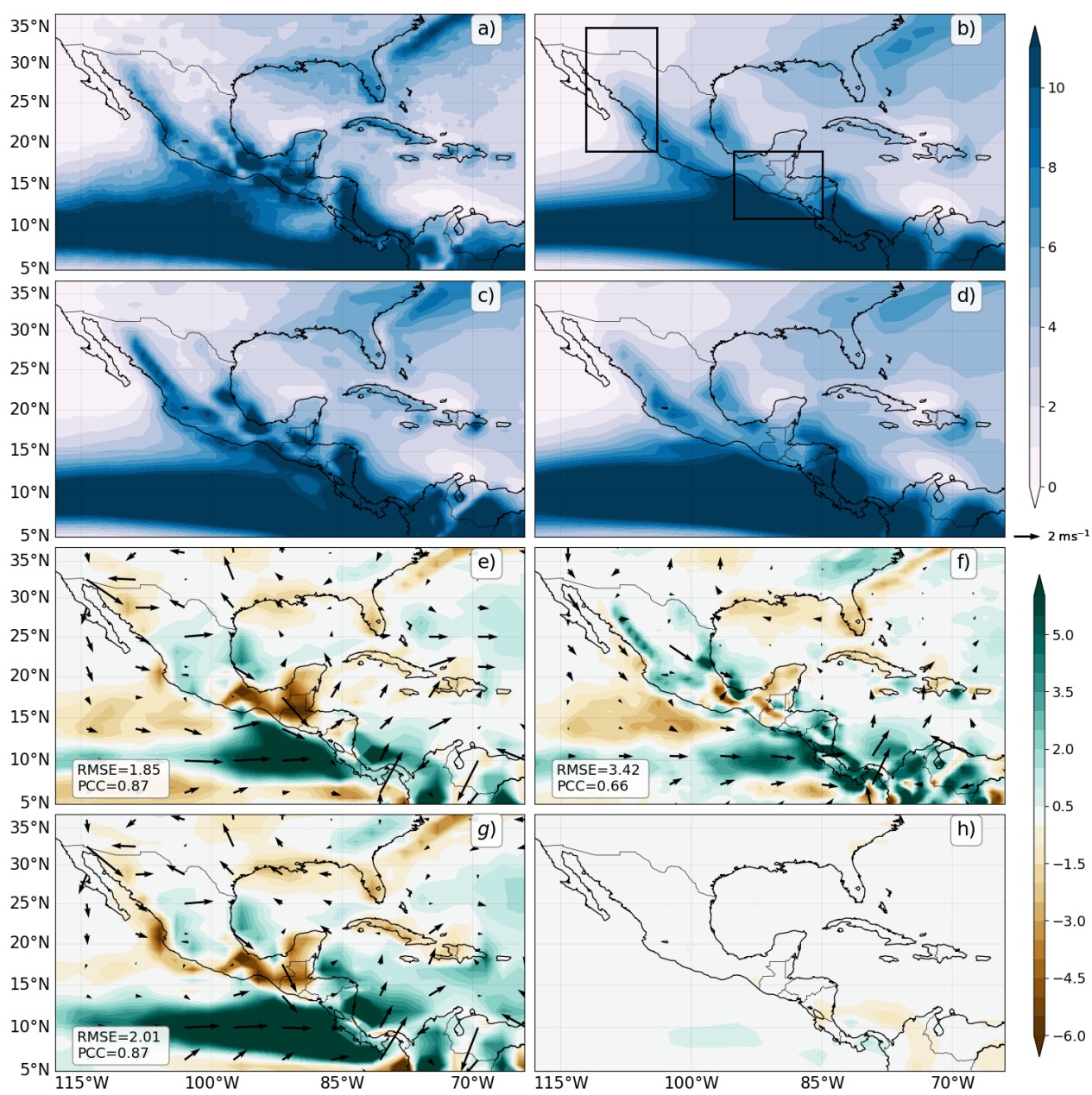

**Figure 7.** As in Figure 6 but for JJA in the northern part of subtropical America.

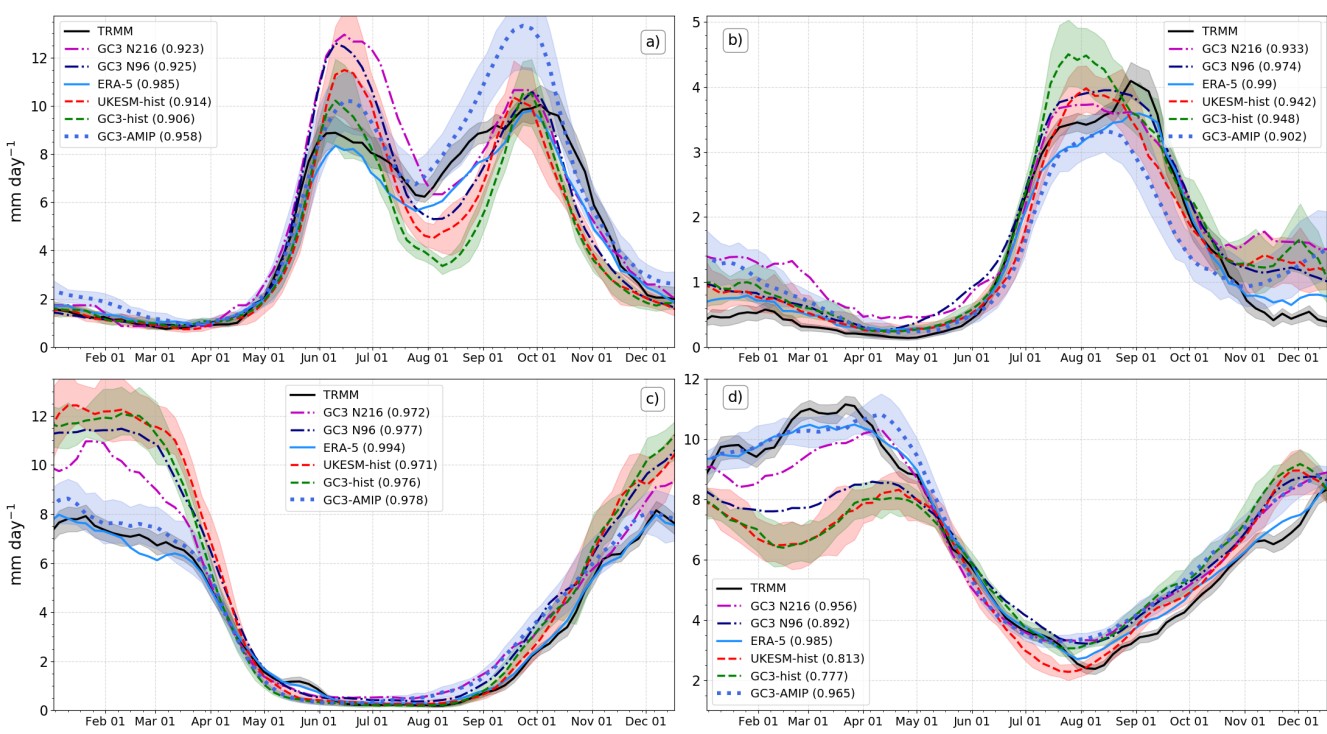

**Figure 8.** Annual cycle of pentad-mean rainfall in the regions (a) the Midsummer drought, (b) the North American Monsoon, (c) Eastern Brazil and (d) the Amazon Basin. The regions are defined as in Figure 3 and are illustrated in Figure 7b and Figure 8b. The shaded regions represent observational uncertainty for TRMM and ensemble spread for the historical experiments. The correlation coefficient for each of the model-driven seasonal cycles with TRMM is given in brackets in each panel.

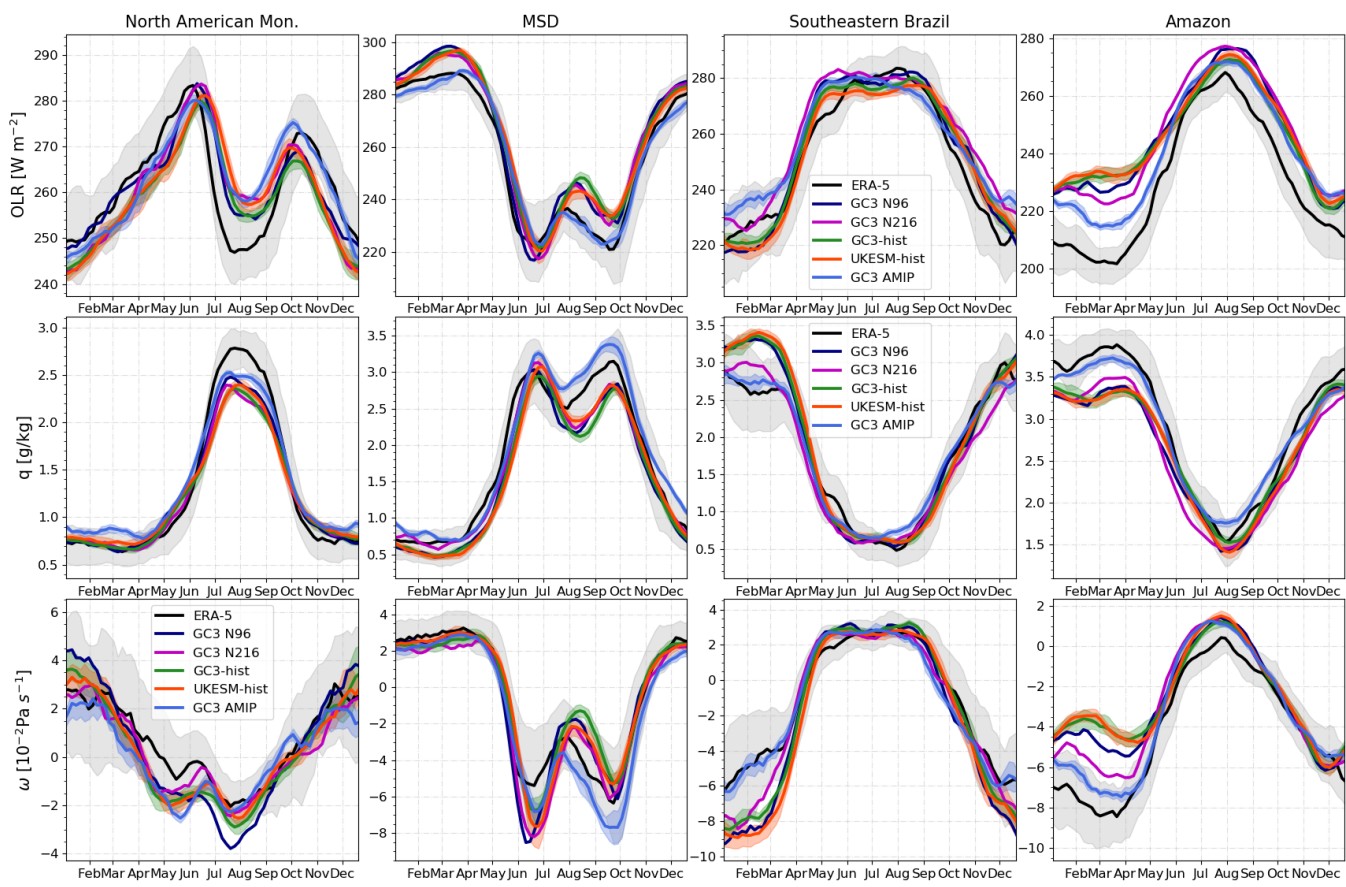

**Figure 9.** Pentad-mean (upper) out-going longwave radiation (OLR), (middle) specific humidity at 500-hPa and (lower) $\omega$ 500-hPa. These are shown from left to right for the North American Monsoon, the Midsummer drought, southastern Brazil and the core Amazon. The uncertainty in ERA-5 data, shown as faint gray shading was estimating by bootstrapping with replacement the ERA-5 record 10,000 times.

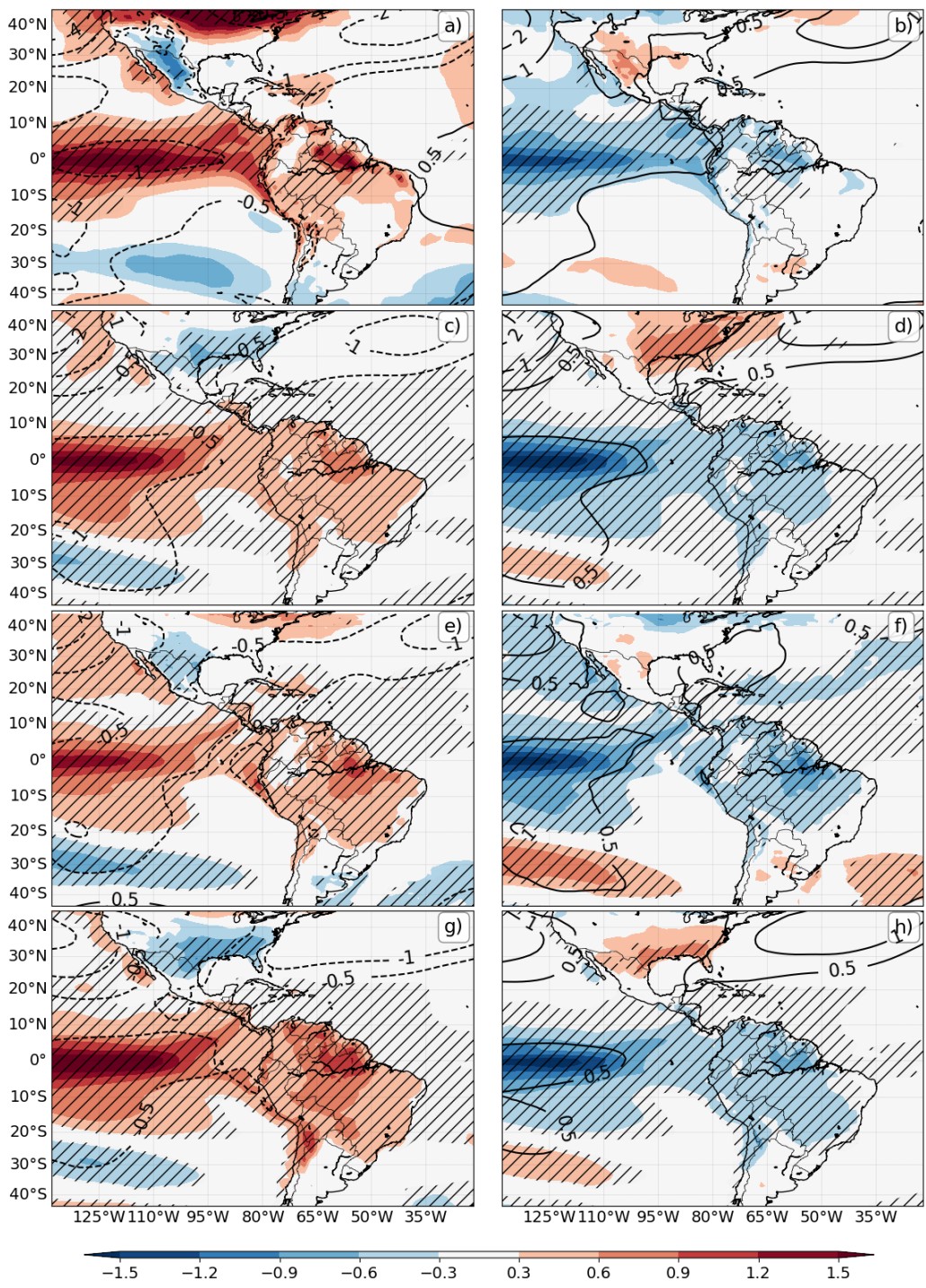

**Figure 10.** DJF Temperature anomalies (colour contours in K) and SLP (line contours in hPa) during (a, c, e, g) El Niño and (b, d, f, h) La Niña events. Results are shown for (a, b) ERA-5, (c, d) UKESM1-historical, (e, f) GC3 N96-pi and (g, h) GC3 N216-pi. The hatched regions denote 99% significance from a Welch t-test for the temperature field.

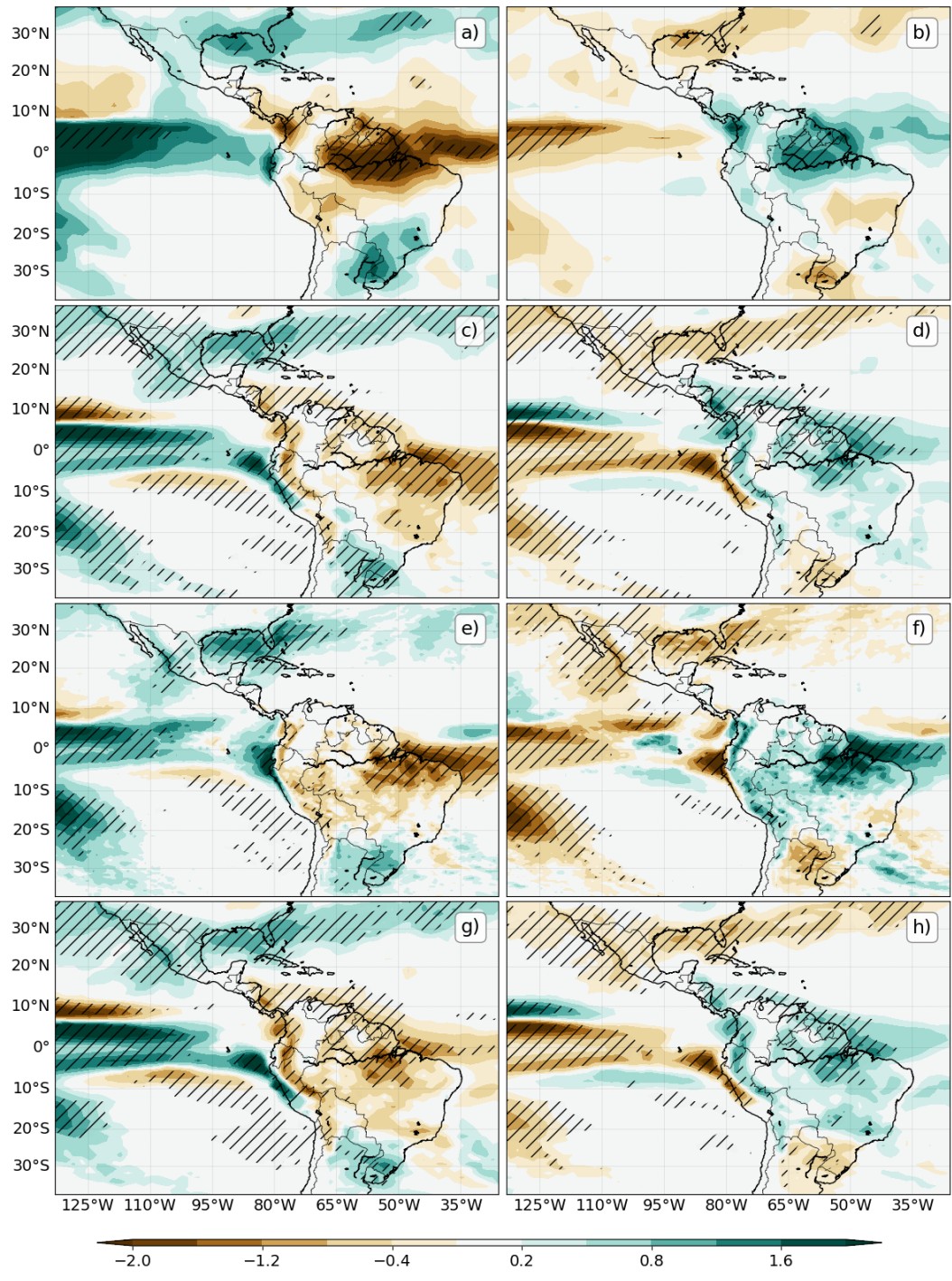

**Figure 11.** As in Figure 10 but for the rainfall response [mm day$^{-1}$] using GPCP as the observational dataset.

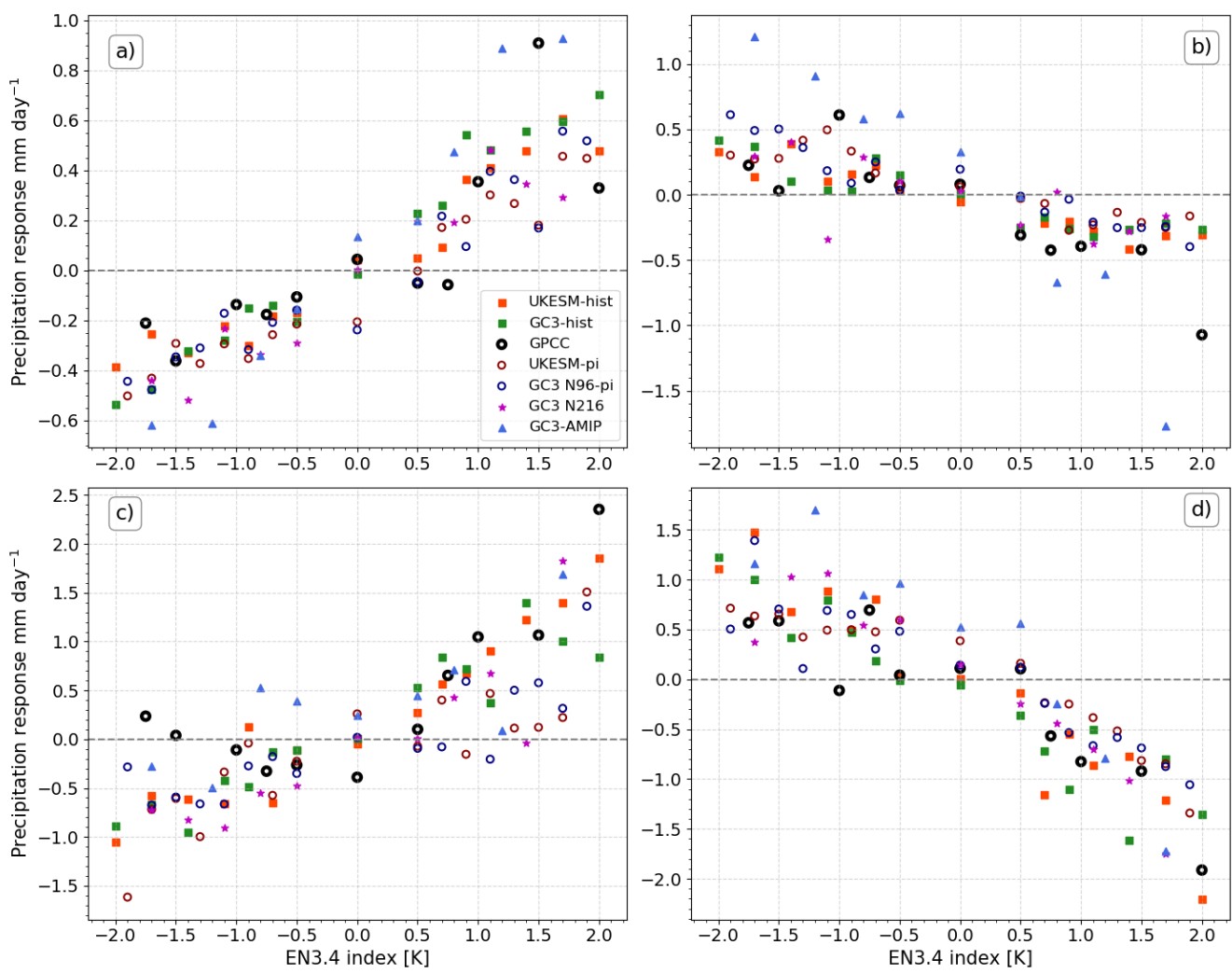

**Figure 12.** Precipitation response [mm day$^{-1}$] as a function of the El Niño 3.4 index (see text) for (a) southwestern North America [20-37°N, 112-98°W], (b) Central America and southern Mexico [5-19°N, 95-83°W],, (c) South Eastern South America [35-25°S, 60-50°W], and (d) the Amazon [10-0°S, 70-45°W]. The observation scatter points are from GPCC in the period of 1940-2013.

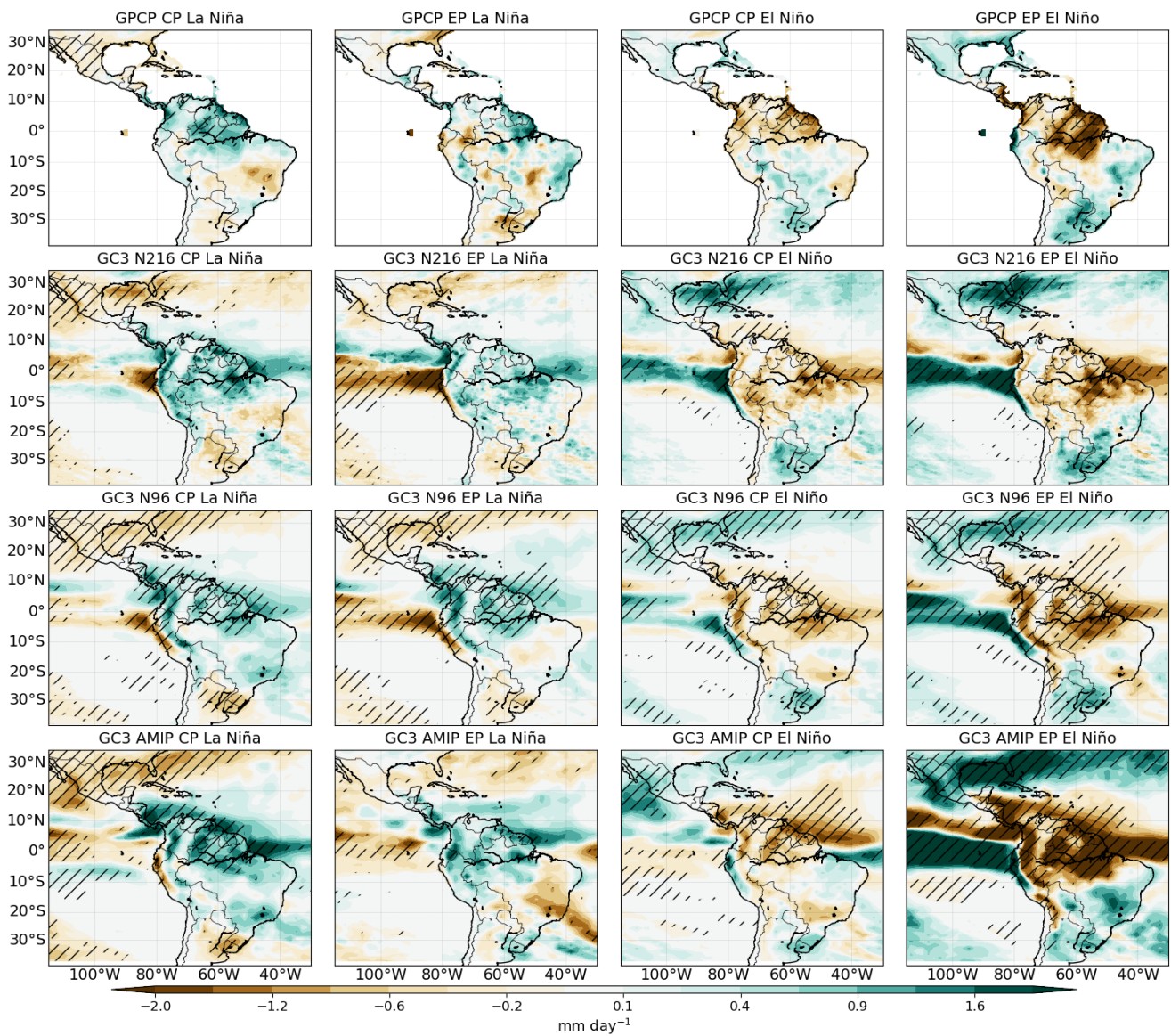

**Figure 13.** Precipitation anomalies in GPCC 1940-2013, GC3 N216, GC3 N96-pi and GC3 AMIP for the four different types of ENSO events, as defined by Cai et al. (2020). Statistically significant anomalies (95% confidence level) are hatched.

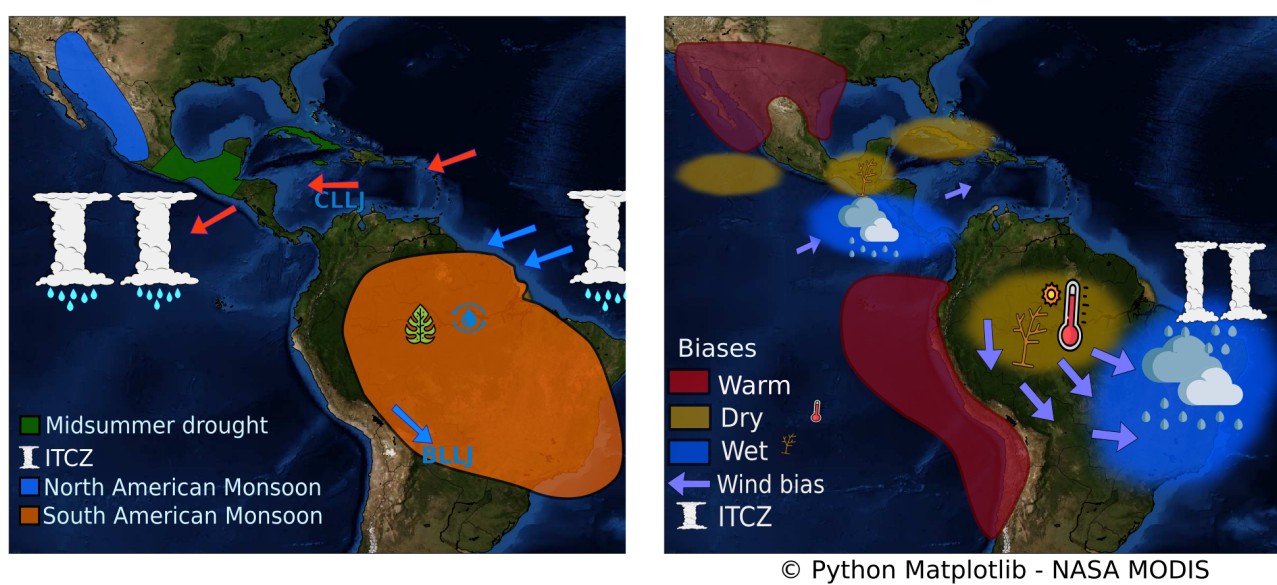

© Python Matplotlib - NASA MODIS

**Figure 14.** Schematics of (left) the main features in the AMS and (right) the main biases in UKESM1 and HadGEM3. In (a) the boreal summer easterlies (red) and austral summer circulation (blue) are shown with the Caribbean and Bolivian Low-level Jets (CLLJ and BLLJ, respectively). In (b) the biases are shown for the respective northern and southern Hemisphere summers. The ITCZ bias in (b) refers to the southward displacement bias of the Atlantic ITCZ in the simulations.

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
