# Peer review of "The American Monsoon System in HadGEM3 and UKESM1"

_Weather and Climate Dynamics, 2020_

## Referee Comment (RC1) · Anonymous Referee #1 · 28 Apr 2020

The submitted manuscript represents a useful contribution to the assessment of the American monsoon systems (both its North and South America components spanning boreal and austral summers) using the latest CMIP-class versions of UK Met Office models. The novelty of the work partially stems from the fact that these monsoon systems are rarely formally assessed in European models such as the UKMO models, and to do so at such an early stage in the CMIP6 lifecycle is a valuable contribution to the literature for this ill-studied monsoon system.

With some corrections the paper should be acceptable for eventual publication in WCD, although it is a matter for the Editors as to whether this paper suits the remit of WCD or would be better served as a model evaluation paper in the sister journal GMD.

Specific comments: The abstract is generally well-designed and describes the work

concisely, although there are a few issues to be considered.

Line 8: The abstract describes that "[the model has] a stronger intraseasonal variation than observed". Note that intraseasonal variability (in the way that most readers will understand the term, i.e. the Madden-Julian Oscillation or Boreal Summer Intraseasonal Oscillation) is not at all examined in this study. The authors are really describing aspects of the annual cycle (e.g. the mid-season drying). Thus, the wording here needs to be changed to avoid the language of intraseasonal variability. (See also later similar comment.)

Line 9: While the Atlantic ITCZ is assessed, what of the SACZ? Is it relevant for such a study of the South American monsoon system?

Line 12: I think it is fair to say that ENSO characteristics (amplitude, frequency, longitudinal position, meridional spread, pattern, skewness...) are not at all assessed in this study. Thus, a more accurate form of words is needed here in order to avoid giving the reader such a misconception, e.g. revised wording should focus on the AMS response to ENSO.

Line 15/16: Instead of "between the two model configurations", in the abstract the sentence should be worded to emphasize the scientific (rather than technical) meaning of this, namely that Earth System processes appear to make no difference to the monsoon simulation.

Line 16: At this stage the various resolutions involved have not been described so the use of the term "medium resolution" may confuse the reader, since it naturally implies there has been a comparison made with both lower and higher resolutions. Being familiar with the model framework used, I know that there is a higher resolution version of the HadGEM3 model although it is not studied here. The authors need to rethink their terminology for this description. (See also later comments.) In addition, the resolutions used need to be explained in the abstract since they mean very different things to different readers (and their definition also changes with as this article ages).

Line 25/26: In an analogous fashion, what about the parts of South America north of the equator and their annual cycle? Can they be aligned to the NAMS?

Line 63: How have CMIP5 models misrepresented the magnitude of the seasonal cycle? Are they systematic under- or over-estimations, or a mixture depending on the model?

Line 67: Be specific as to what the CMIP5 models have improved upon. Presumably it is the CMIP3 models.

Table 1: Which version of the TRMM algorithm is used? V7 for example is known to perform better over orography such as the Andes (see Zulkafli et al., 2014, https://doi.org/10.1175/JHM-D-13-094.1).

Line 96/97: What is the evidence that TRMM provides the most reliable source of information on rainfall for this region? Are there citable studies intercomparing satellite, gauge and merged datasets for the NAMS and/or SAMS?

Lines 100-124: Note that the ocean model horizontal resolutions have not been listed.

Line 129: Clarify whether this is surface temperature or surface-air (i.e. 1.5 or 2 metre) temperature that is being considered.

Line 132: The Welch t-test should be defined in the methods or referenced here. How does this differ from a student's t-test? Plus, how are the different ensemble members dealt with relative to this?

Line 137/138: Does the stronger Bolivian LLJ support a stronger seasonal cycle/monsoon in the region north of the equator in South America during boreal summer? (I.e. in the South America component of the NAMS.)

Line 147/148: The physical outcome of this needs to be made explicitly clear to the reader, namely it appears that the inclusion of Earth System processes makes no difference to the SAMS.

[Figure]

Line 149/150: A better summary of the changes in historical forcing (compared to the pre-industrial) needs to be described in lines 119-124 in order for the reader to be able to understand possible changes. Clearly the reader will know that global GHG emissions have increase, but what are the relevant/local patterns of aerosol emissions, land-use change etc. between the two experiments?

Line 150-152: Given the length of the pre-industrial control integrations that are available (and given the small size of the forcing when compared to the historical experiment), the internal variability of quantities such as those listed here (and elsewhere through the results) within the pre-industrial should be considered as a means to understanding the significance of any change.

Line 175: I understand the logic, but the chosen model comparison mixing UKESM with the GC3 model appears rather unclean.

Line 193: In what way is the low-level wind structure biased?

Lines 171-222 and onwards: All of the comparisons whether maps or seasonal cycles would benefit from a table of quantitative comparisons between the various datasets, such as pattern correlations (or just correlations for the seasonal cycles) and RMSE. This is standard practice in multi-model evaluation studies.

Line 239: That the AMIP models "removed the spatial patterns" is strange wording. Did any bias remain at all? Generally, I think that this study could be significantly strengthened if a fuller comparison could be made between AMIP runs of these two models (which will be available as contributions to the CMIP6 DECK) could be thoroughly compared with the coupled historical runs. The absence of SST bias would make for improved understanding.

Line 253: Here the run is referred to as "high-resolution" yet in the abstract it was medium resolution. The consistency within the manuscript needs to be improved. Could the manuscript not also examine a higher resolution version of the GC3 experiment, e.g. at N512?

Line 262: Are there any published onset measures for the AMS that could be used to measure this? And how is the onset objectively defined from Figure 9b? E.g. 1mm/d threshold, or the maximum rate etc.?

Lines 256-287: In the tropics, and especially for monsoons, I would expect the seasonal cycle of precipitation to be discussed in the context of the lower tropospheric circulation. This doesn't necessarily need to be done in the same paragraphs (the layout here is fine), but at the very least I would expect the discussions of precipitation biases here to reference the circulation biases for consistency. This is because of the intimate connection between circulation and precipitation in the tropics: winds providing moisture to the monsoon and the monsoon heating feeding back on the circulation to bring more moisture. At present the discussion is kept very separate. This could be aided by adding wind vectors to Figures 7 and 8.

Lines 256-287: It would be preferable to have some contextual comparison with other contemporary models (or at least CMIP5). How did CMIP5 perform for the NAMS and SAMS (cite references)? Do the UKMO models here fit within that envelope or are they better/worse? This will help improve the level of interest in this study outside the single modelling group. Furthermore, can the authors state how the current UKMO model versions (especially GC3.1) have advanced upon earlier versions (HadGEM3, HadGEM2-ES, even HadCM3) with respect to the AMS? Are there any published works mentioning those models? It would be useful for the community to understand if the simulation is being improved or whether significant biases are persisting.

Line 288: In the deep tropics, OLR is not really going to tell us much more than we already learn from precipitation, since much of the convection is deep. What is the nature of convection in the regions discussed? If any particular regions are dominated by shallow convection/warm rain, then this could be highlighted by references to relevant published works.

Line 297: How certain can we be about the tropospheric moisture in any case in a reanalysis? What level of data is assimilated in some of these remote regions? Can any ground-truthing (really air-truthing!) be performed (even if not shown) using nearby RS launches such as those publicly available from Wyoming?

Line 328: Unlike the implication in the abstract, there is no assessment made here of general ENSO behaviour in these coupled models – and if the driving point of a teleconnection is faulty then resultant impacts over the AMS will hardly do well. A summary of the behaviour of ENSO in these models with reference to a published assessment of their performance should be made.

Line 332: Is this in units of temperature (degC/K) or a normalised index in terms of standard deviations? Where is the index taken from or how have you calculated it?

Line 334: What are the years included in the observed composites of El Nino and La Nina? Has the impact of CP and EP El Nino events been considered and what does the published literature say about the different impacts of such events on the NAMS and SAMS?

Line 351: It would be very instructive if wind vectors were added to Figure 12, enabling to reader to understand something of the mechanism by which ENSO controls rainfall anomalies in the AMS. The authors should then elaborate upon this in the text.

Line 348-350: It's not immediately obvious how the NAO links described are relevant to the study at hand. The authors should either make this clear or remove this text.

Lines 365-370: The authors should consider whether the lack of nonlinearity in the modelled ENSO response reflects the lack of diversity of simulated ENSO in the model (e.g. the lack of distinct central Pacific or east Pacific events).

Line 376: The authors could be more explicit on the likely kink between cloud cover and the warm bias in the SAMS domain. If precipitation is too weak, this should be stated explicitly. (Note there would also be a soil-moisture feedback as a result.)

[Figure]

Lines 376-380: Finish the sentence by making explicit how the land-sea temperature contrast may feedback on the monsoon.

Line 391: Make explicit whether the Ryu and Hayhoe study was using CMIP5 models.

Line 393: With reference to the earlier comment on the abstract, the authors should avoid the terminology of intraseasonal variability here since the MJO/BSISO have not been assessed.

Lines 371-404: In the conclusions I would want to see a more thorough synthesis of the results (e.g. how all the meteorological components fit together) than a summary of each in turn. It would also be worth reflecting upon (if possible) how these models sit in comparison to published literature on the AMS in CMIP3/5 models or on earlier versions of UKMO models.

Line 413: See earlier comments on higher/medium resolution.

Line 418: Need to see a summary of how the Earth System processes influence the response to forcing.

Figure 1: The domains used later in Figure 3 etc. need to be pictured somewhere, e.g. on this figure.

Trivia: Lines 13/14: Perhaps replace "in subtropical America" [meaning USA?] with "in the subtropical Americas".

Line 21: Change "copuled" to "coupled".

Line 42: "...and the dynamics the features largely characterise the MSD characteristics...". I don't understand what is meant here, something is wrong with the grammar.

Line 43: Change "reproduce accurately" to "accurately reproduce".

Line 51: Remove hyphen from "South-America".

Line 66: Space needed in "MetOffice".

Line 119: Replace "beginning for" with "covering"; replace "that include" with "of".

Line 142: Change "temperature" to "temperatures".

Line 171: Second "the" is not required.

Line 184: brackets not needed around location point.

Line 195: By "a minimum" do you mean "southernmost position"? This would be easier to understand.

Line 302: Replace "indicative" with "are indicative".

Line 304: Clarify if the decreased omega is a reduction or increase in ascent.

Line 309: Mixture of singular and plural in this line.

Line 331: By convention, "El" is not included when referring to the "Nino-3.4 index".

Line 362: Change "opposite sign response" to "opposite signed response".

---

## Referee Comment (RC2) · David Adams (Referee) · 2 May 2020

Review of *The American Monsoon System in HadGEM3.0 and UKESM1CMIP6 simulations*
 by Garcia-Franco et al.

David K. Adams
dave.k.adams@gmail.com

Recommendation:  Minor Revision

**Major Comments**
The authors have presented a well-written manuscript on the important topic of analyzing bias in Global Climate Models in their transition to Earth System Models with coupling to the land/biosphere, ocean and sea ice and well as chemical cycles and aerosol interactions.   In this case, the focus is on the American Monsoon System in the context of the HadGEM3.0 GC3.1 and the UKESM1 models.  The basic presentation is clear and indicates where biases in the distribution and intensity of, for example, precipitation are found across the Americas.  Nevertheless,  I do have two particular criticisms of the manuscript: (1) there should be a little context given in terms of moving towards high resolution (towards kilometer scale) ESMs  and,  (2) some attempt at an explanation, even if speculative, as to why the biases and differences are observed in the model results (e.g., different physical parameterizations).

With respect to criticism (1), I think you could expand a little bit on the movement towards higher resolution ESM models that has arisen in the last few years.  This would help better motivate the work done in this study.   For example, these studies and commentaries  (Stevens and Bony 2013; Marotzke et al., 2017; Schulthess et al., 2018; Palmer and Stevens, 2019; Stevens et al., 2019) could be relevant to better motivating this study. And (2) also related to the above mentioned studies in several cases, is to indicate which physical parameterizations in the model could be most associated with the model biases.  Clearly, the spatial and temporal distribution of rainfall will be related to model convective and microphysical parameterizations and would be obvious choices to consider. Likewise, these parameterizations determine the distribution of rainfall in the tropics and associated latent heat release and are, therefore, fundamental to the intensity of large-scale circulation features such as the ITCZ and the displacement and intensity of the sub tropical high pressure systems, etc …

**Minor Comments**

Line 36 " A bimodal regime characterises the seasonal cycle of precipitation in southern Mexico, Central America and the Caribbean  that is typically referred to as Midsummer Drought (MSD)"
Perhaps for completeness you can include the more local reference terms for this phenomenon.
In Central America it is often called "El Veranillo"  and in southern and eastern Mexico "La Canícula".
For example a bit more detail on the MSD can be found in these articles, Amador JA et al. 2016, Amador, J.A., et al., (2016),  Durán Quesada et al (2017).

Line 41 . The complex  interplay of moisture transport, evaporation and the dynamics..."
When you say evaporation here you should probably clarify if you mean from the sea-surface or from land-surface or both, as terrestrial latent heat fluxes is a difficult quantity to measure and the effects on precipitation are unclear (e.g., moisture recycling).

Line 51 "The date of monsoon onset is also region-dependent; in northern South-America convection is observed from early October, whereas convection in southeastern Brazil typically starts in mid-November or later (Marengo et al., 2001; Nieto-Ferreira and Rickenbach, 2011)."
You probably want to clarify this. Do you mean deep convection and the associated rainy season?
In the Central Amazon region, the rainy season begins late December and lasts until about April. Typically, in October, there may be intense deep convective events in the Central Amazon (see Adams et al 2013), but in terms of convective precipitation, January through April are very rainy (see Machado et al. 2004). How well models actually reproduce the geographic distribution of Amazon Basin rainfall is an important issue, you may want to discuss with a little more detail and citations.

Line 69 You should probably clarify what hemisphere you mean here when you say "fall".

Line 73 "The next efforts to improve climate models include increased horizontal resolution, ..."
This drive towards increased horizontal resolution is quite strong, down to the kilometer resolution for GCMs, you should refer to some of the literature I mention in the Major Comments section.

Line 83 "The study documents the main biases in the simulated climate of UKESM1 and HadGEM3.0 and compares the effect of increased horizontal resolution and Earth System processes on the representation of the AMS climate. The analysis provides a framework for using these climate models in scenario studies, to highlight possible sources of model error that may be corrected and to further understand variability and teleconnections in this region."

Line 98 "GPCP, GPCC and CHIRPS are also used for their longer period, although arguably each of these datasets have shortcomings in either resolution or spatial coverage."
You should probably include a few citation of studies that have used these data in similar context for the reader to consider, particularly studies where the shortcomings are discussed.

Line 113 "piControl", I assume you mean pre-industrial, but you should spell it out for the reader.

Line 190 "Afterwards, the ITCZ migrates northward reaching a peak latitude and mean rainfall at 10 ∘ N by day 250, or May 30." I think you have made a mistake here, you probably mean early September.

Line 213 Write "Negative ω and low-level moisture biases in the central and East Pacific Ocean ..."

Line 221 "These are observed as negative zonal wind biases, indicative of significantly weaker upper-level westerlies resulting from the overturning circulation in the Pacific Ocean."
This statement is a bit confusing, it sounds as if you are referring to the oceanic circulation within the Pacific Ocean.

Line 280 Rewrite using commas "The models also show a good representation of the transition from winter to summertime rainfall by representing, with relative skill, the smooth transition from 4 mm day −1 in September to 6 mm day −1 in November and close to 8 mm day −1 in late December."

Line 290 "these quantities characterise the strength and height of deep convection and the mid-level moisture." This idea is a little unclear, what you do mean "the mid-level moisture"? Specific humidity has a vertical distribution associated with instability and convection. And OLR for convective cloudiness would be associated with high levels in the atmosphere.

Line 319   Check spelling  "althougth"

**References**

Adams, D. K.,  S. I. Gutman, K. L. Holub, and D. S. Pereira, 2013: GNSS observations of deep convective time scales in the Amazon. Geophys. Res. Lett., 40, 2818–2823, doi:10.1002/grl.50573.

Amador JA, Rivera ER, Durán-Quesada AM, Mora G, Sáenz F, Calderón B, Mora N. The easternmost tropical Pacific. Part I: A climate review. Revista de Biología Tropical. 2016 Mar 2:1-22.

Amador, J.A., Durán-Quesada, A.M., Rivera, E.R., Mora, G., Sáenz, F., Calderón, B. and Mora, N., 2016. The easternmost tropical Pacific. Part II: Seasonal and intraseasonal modes of atmospheric variability. *Revista de Biología Tropical*, pp.23-57.

Durán Quesada, Ana María, Luis Gimeno, and Jorge Alberto Amador Astúa. "Role of moisture transport for Central American precipitation." (2017).

Machado, L. A. T., H. Laurent, N. Dessay, and I. Miranda, 2004: Seasonal and diurnal variability of convection over the Amazonia: A comparison of different vegetation types and large scale forcing. Theor. Appl.Climatol., 78, 61–77, doi:10.1007/s00704-004-0044-9.

Marotzke, J., C. Jakob, S. Bony, P. A. Dirmeyer, P. A. O'Gorman, E. Hawkins, S. Perkins-Kirkpatrick, C. Le Quéré, S. Nowicki, K. Paulavets, S. I. Seneviratne, B. Stevens, and M. Tuma, (2017), Climate research must sharpen its view, Nature Climate Change, 7, 89-91.

Schulthess, T.C., P. Bauer, N. Wedi, O. Fuhrer, T. Hoefler and C. Schär, (2019), Reflecting on the Goal and Baseline for Exascale Computing: A Roadmap Based on Weather and Climate Simulations,Computing in Science &Engineering, 21,30-41. https://doi.org/10.1109/MCSE.2018.2888788

Sherwood, S. C., S. Bony, and J.-L. Dufresne (2014), Spread in model climate sensitivity traced to atmospheric convective mixing. Nature, 505 (7481), 37

Stevens, B., S. Bony (2013), What Are Climate Models Missing, Science, 340, 6136, 1053-1054, doi: 10.1126/science.1237554

Stevens, B., Satoh, M., Auger, L. et al. (2019), DYAMOND: the DYnamics of the Atmospheric general circulation Modeled On Non-hydrostatic Domains, Prog. Earth Planet. Sci., 6, 61, doi:10.1186/s40645-019-0304-z.

---

## Author Comment (AC1) · 3 Jun 2020

**Response to Referee #2**

García-Franco, JL., Gray LJ, Osprey S.

*Atmospheric, Oceanic and Planetary Physics, Department of Physics, University of Oxford. Parks Road Oxford, United Kingdom OX1 3PU email: jorge.garcia-franco@physics.ox.ac.uk*

Dear Dr. David Adams,

Many thanks for your comments and suggestions on the manuscript. We have found your two main critiques to be very timely and useful to improve our study. We hope you'll find the changes made to manuscript satisfactory.

Regarding your two major comments we have made the following changes to the manuscript. First, we included the following in the introduction to motivate the study:

Climate research in recent decades has aimed to reduce uncertainty in climate projections by improving GCMs, but different approaches taken by modelling centres appear to be seemingly disconnected (Jakob, 2014). One approach is to reduce horizontal resolution down to km resolution to rely less on parametrizations and more on physical laws to represent clouds and convection (Palmer and Stevens, 2019). A second approach aims to model Earth System processes to better characterise complex land-atmosphere-ocean biogeochemical cycles that may provide a better constraint on climate sensitivity, a parameter that depends on the carbon cycle (Marotzke et al., 2017; Sellar et al., 2019; Andrews et al., 2019). Finally, recent arguments have also suggested to include stochastic parametrisations of sub-grid processes since this approach has improved seasonal forecasts and may therefore improve climate projections (Palmer, 2019).

To address your second comment regarding perhaps the parametrisations and their role for improved biases, as we mostly see an improvement of the biases in the South American Monsoon with increased resolution we argue that the oceanic component is very relevant, where perhaps the eddy heat flux parametrisations improve as resolution improves.

The results of this study showed that the medium resolution (GC3 N216) simulation improved upon some of the biases of the lower resolution simulations, such as most of the precipitation biases. This improvement in the medium resolution simulation may largely be due to the improved dynamics associated with relying less on model parametrisations and more on physical governing laws. The double-ITCZ problem and the Atlantic ITCZ biases have been shown to be directly affected by the convective scheme. Several parametrised processes associated with the convective scheme have been shown to the treatment of entraintment and moisture-cloud feedbacks (Oueslati and Bellon, 2013; Li and Xie, 2014). The resolution of the ocean sub-model is known to have an impact over the equatorial Atlantic SSTs and the ITCZ biases which are noticeably reduced in the medium resolution simulation (Kuhlbrodt et al., 2018), due to the improvement in the eddy heat flux and the 15associated heat uptake and transport of the ocean. The

improvement in the ITCZ and the associated dynamics also improves the associated circulation biases in the South American Monsoon indicating that the oceanic resolution in these models improve the cross-equatorial transport, SST gradients and the land-sea circulation over the Amazon during austral summer.

**1. Specific comments**

Line 36 " A bimodal regime characterises the seasonal cycle of precipitation in southern Mexico, Central America and the Caribbean that is typically referred to as Midsummer Drought (MSD)" Perhaps for completeness you can include the more local reference terms for this phenomenon. In Central America it is often called "El Veranillo" and in southern and eastern Mexico "La Canícula". For example a bit more detail on the MSD can be found in these articles, Amador JA et al. 2016, Amador, J.A., et al., (2016), Durán Quesada et al (2017).

We have added the suggested references and more detail to this paragraph:

Line 41 . The complex interplay of moisture transport, evaporation and the dynamics..." When you say evaporation here you should probably clarify if you mean from the sea-surface or from land-surface or both, as terrestrial latent heat fluxes is a difficult quantity to measure and the effects on precipitation are unclear (e.g., moisture recycling).

The manuscript now states:

The complex interplay of SSTs, evaporation and moisture between the East Pacific Ocean and the Caribbean Sea are key for the spatial and temporal characteristics of the MSD (Amador et al., 2006; Herrera et al., 2015; Durán-Quesada et al., 2017; Straffon et al., 2019)

Line 51 "The date of monsoon onset is also region-dependent; in northern South-America convection is observed from early October, whereas convection in southeastern Brazil typically starts in mid- November or later (Marengo et al., 2001; Nieto-Ferreira and Rickenbach, 2011)." You probably want to clarify this. Do you mean deep convection and the associated rainy season? In the Central Amazon region, the rainy season begins late December and lasts until about April. Typically, in October, there may be intense deep convective events in the Central Amazon (see Adams et al 2013), but in terms of convective precipitation, January through April are very rainy (see Machado et al. 2004). How well models actually reproduce the geographic distribution of Amazon Basin rainfall is an important issue, you may want to discuss with a little more detail and citations.

Many thanks for these suggestions, the initial wording was indeed unclear. We hope you like the new wording in the manuscript. Now, when presenting the South American Monsoon:

In the central Amazon and northern South America, convective activity is observed from early October but the main rainy season extends from December to April (Machado et al., 2004; Adams et al., 2013), whereas convection in southeastern Brazil starts in November and peaks in January and February (Marengo et al., 2001; Nieto-Ferreira and Rickenbach, 2011).

Line 69 You should probably clarify what hemisphere you mean here when you say "fall".

*Done, we clarified to state "austral fall".*

Line 73 "The next efforts to improve climate models include increased horizontal resolution, ..." This drive towards increased horizontal resolution is quite strong, down to the kilometer resolution for GCMs, you should refer to some of the literature I mention in the Major Comments section.

*We have addressed this from your major comment.*

Line 83 "The study documents the main biases in the simulated climate of UKESM1 and HadGEM3.0 and compares the effect of increased horizontal resolution and Earth System processes on the representation of the AMS climate. The analysis provides a framework for using these climate models in scenario studies, to highlight possible sources of model error that may be corrected and to further understand variability and teleconnections in this region."

Line 98 "GPCP, GPCC and CHIRPS are also used for their longer period, although arguably each of these datasets have shortcomings in either resolution or spatial coverage." You should probably include a few citation of studies that have used these data in similar context for the reader to consider, particularly studies where the shortcomings are discussed.

*Reviewer 1 made a similar suggestion. Therefore, now we point to studies that have validated one or several or these datasets in a region of the AMS. However, a study intercomparing the different datasets and validating them against rain gauge data is not know to us. The manuscript now reads:*

*Some studies have validated one or several of these datasets in the AMS region (e.g. Franchito et al., 2009; Dinku et al., 2010; Trejo et al., 2016).*

Line 113 "piControl", I assume you mean pre-industrial, but you should spell it out for the reader.

*Done.*

Line 190 "Afterwards, the ITCZ migrates northward reaching a peak latitude and mean rainfall at 10N by day 250, or May 30." I think you have made a mistake here, you probably mean early September.

*Correct. We corrected this line.*

Line 213 Write "Negative $\omega$ and low-level moisture biases in the central and East Pacific Ocean ..."

*Done.*

Line 221 "These are observed as negative zonal wind biases, indicative of significantly weaker upper-level westerlies resulting from the overturning circulation in the Pacific Ocean." This statement is a bit confusing, it sounds as if you are referring to the oceanic circulation within the Pacific Ocean.

*This statement has now been removed as the Walker circulation figure has been moved to the Supplementary material and the discussion of the figure in the main paper has been made shorter.*

Line 280 Rewrite using commas "The models also show a good representation of the transition from winter to summertime rainfall by representing, with relative skill, the smooth transition from 4 mm day-1 in September to 6 mm day -1 in November and close to 8 mm day -1 in late December."

*Done.*

Line 290 "these quantities characterise the strength and height of deep convection and the mid-level

moisture." This idea is a little unclear, what you do mean "the mid-level moisture"? Specific humidity has a vertical distribution associated with instability and convection. And OLR for convective cloudiness would be associated with high levels in the atmosphere.

Yes, the wording in this paragraph was unclear. We use the free tropospheric specific humidity values ($q$ at 500 hPa) and the vertical velocity at the same level, as well as OLR. And yes, these are technically different levels in the atmosphere we are looking at, and only a small picture of the vertical characteristics of convection, however, these metrics show interesting differences between model and reanalysis data.

Line 319 Check spelling "althougth"

Done.

---

## Author Comment (AC2) · 3 Jun 2020

**Response to Referee #1**

García-Franco, JL., Gray LJ, Osprey S.

*Atmospheric, Oceanic and Planetary Physics, Department of Physics, University of Oxford. Parks Road Oxford, United Kingdom OX1 3PU email: jorge.garcia-franco@physics.ox.ac.uk*

We thank Reviewer 1 for the positive and constructive comments on the manuscript. We agree that this analysis is valuable for understanding the model performance of the UKMO models in a relatively less studied monsoon region. We now address each of your comments (in blue), highlighting the changes done to the manuscript as a result of your comment (in purple). In short, we have identified two major concerns that you presented and addressed them by including the AMIP simulation in all the analyses, to better understand the role of SSTs biases, as well as further analysing ENSO characteristics and the impact of ENSO diversity on the teleconnections to these monsoon systems by including a figure in the manuscript showing the different simulated and observed responses to the different types of ENSO events.

**1. Specific comments**

Line 8: The abstract describes that "[the model has] a stronger intraseasonal variation than observed". Note that intraseasonal variability (in the way that most readers will understand the term, i.e. the Madden-Julian Oscillation or Boreal Summer IntraseasonalOscillation) is not at all examined in this study. The authors are really describing aspects of the annual cycle (e.g. the mid-season drying). Thus, the wording here needs to be changed to avoid the language of intraseasonal variability. (See also later similar comment.)

The revised manuscript has changed the language to specify that these models have a stronger difference between the two peaks of precipitation and the mid-season dry period.

Line 9: While the Atlantic ITCZ is assessed, what of the SACZ? Is it relevant for sucha study of the South American monsoon system?

The SACZ is very relevant as a major driver of variability in precipitation and circulation in the SAMS. We have addressed your comment by adding a supplementary figure comparing the modelled and simulated SACZ spatial patterns and seasonal cycles, although the abstract makes no mention of this. This figure is now mentioned in the manuscript in section 3.2 as follows:

The South Atlantic Convergence Zone (SACZ) is a nortwest-southeast oriented band of convection and is a prominent influence on the South American Monsoon mean and extreme rainfall (Carvalho et al., 2004; Marengo et al., 2012). UKESM1 and GC3 appear to reasonably simulate the spatial pattern of active SACZ days and the seasonal cycle of SACZ activity (Figure S2).

Line 12: I think it is fair to say that ENSO characteristics (amplitude, frequency, longitu-dinal position, meridional spread, pattern, skewness...) are not at all assessed in this study. Thus, a more accurate form of

words is needed here in order to avoid giving thereader such a misconception, e.g. revised wording should focus on the AMS responseto ENSO.

The wording has been changed to focus on the teleconnections and the response of the AMS to ENSO. However, the characteristics of ENSO in the models, such as those mentioned by the reviewer are in good agreement with observations. For example, Menary et al. (2018) showed that the power spectrum of ENSO agrees better with the observed HadSST than most CMIP5 and CMIP3 models. We discuss this further in a reply to another comment below.

Line 15/16: Instead of "between the two model configurations", in the abstract the sentence should be worded to emphasize the scientific (rather than technical) meaning of this, namely that Earth System processes appear to make no difference to the monsoon simulation.

Done.

Line 16: At this stage the various resolutions involved have not been described so the use of the term "medium resolution" may confuse the reader, since it naturally implies there has been a comparison made with both lower and higher resolutions. Being familiar with the model framework used, I know that there is a higher resolutionversion of the HadGEM3 model although it is not studied here. The authors need torethink their terminology for this description. (See also later comments.) In addition,the resolutions used need to be explained in the abstract since they mean very different things to different readers (and their definition also changes with as this article ages).

We now use the terminology used in previous papers (e.g Menary et al., 2018) using the HadGEM3 model at N96 and N216 resolution, referring to these runs as low and medium-resolution. The term "high-resolution" is now erased from the manuscript as to avoid possible confusion with higher resolution versions of the MOHC models.

Line 25/26: In an analogous fashion, what about the parts of South America north of the equator and their annual cycle? Can they be aligned to the NAMS?

While parts of southern Central America, e.g., Panama and northern South America, e.g., Colombia, may fit the definition of global monsoon, characterised by a strong seasonal contrast in precipitation, the regions experience a slightly different seasonal cycle, as they are transition zones between the North and South American monsoon with significant rainfall rates in the fall and spring season. Perhaps for this, or other historic reasons, these regions are excluded from North and South American Monsoon literature (Adams and Comrie, 1997, Arritt et al., 2000, Vera et al., 2006, Jones and Carvalho, 2013, Geil et al., 2013), and they are rarely discussed in this manuscript. However, the manuscript does clarify this by stating that the manuscript uses the standard definitions for the North and South American Monsoon, and additionally the MSD region. The manuscript now reads:

This study uses the definitions for the North and South American Monsoons as previous studies (Vera et al., 2006; Marengo et al., 2012), with additional analysis on the MSD of southern Mexico and Central America (Magaña et al., 1999; Perdigón-Morales et al., 2018).

Line 63: How have CMIP5 models misrepresented the magnitude of the seasonalcycle? Are they systematic under- or over-estimations, or a mixture depending on themodel?

The manuscript now clarifies this point. CMIP5 models showed an earlier, but clearly observed, onset date, but the retreat date was unclear from precipitation time-series, due to problems in simulating the large and regional-scale processes associated with the retreat, i.e., the displacement of the subtropical jet. We have thus changed this paragraph as follows:

The CMIP5 simulations of the North American Monsoon misrepresented aspects of the seasonal cycle of precipitation and overestimated the peak monsoon rainfall (Geil et al., 2013; Sheffield et al., 2013a). Most CMIP5 models simulated an earlier onset date, but improved from CMIP3 since the onset date showed a clear separation of rainy and dry seasons in daily precipitation time-series. In contrast, the simulated retreat date was unclear in most models which highlighted problems for these models to simulate the regional changes during retreat stage (Geil et al., 2013; Sheffield et al., 2013a).

Line 67: Be specific as to what the CMIP5 models have improved upon. Presumably itis the CMIP3 models. A more specific and detailed description of biases in the CMIP5 cohort for the 3 regions is given.

Table 1: Which version of the TRMM algorithm is used? V7 for example is knownto perform better over orography such as the Andes (see Zulkafli et al., 2014,https://doi.org/10.1175/JHM-D-13-094.1).

We use the product from satellite 3B42, algorithm V7, the table now clarifies the version used.

Line 96/97: What is the evidence that TRMM provides the most reliable source of information on rainfall for this region? Are there citable studies intercomparing satellite,gauge and merged datasets for the NAMS and/or SAMS?

There are scarce studies that compare precipitation datasets in these regions and in-situ ground based stations managed by local governments have a short record or calibration problems, therefore there is hardly evidence to support the original statement of the manuscript. As the original sentence then lacked support from the evidence, the new manuscript now simply argues that TRMM is used as it has been considered by literature (cited in the manuscript) as a reliable source of information about the spatial and temporal characteristics of precipitation and therefore typically used in GCM evaluation in this region. Furthermore, we do point the reader to relevant comparison studies over several regions of the AMS as follows:

The TRMM dataset has a high horizontal and temporal resolution and was used in several CMIP assessments (Geil et al., 2013; Jones and Carvalho, 2013) as a reliable source of precipitation (Carvalho et al., 2012). Therefore, we use TRMM as our best estimate for the spatial and temporal characteristics of the AMS rainfall. However, the period covered by TRMM (1998-2018) is too short to analyse statistically robust teleconnections or variability, so we use GPCP, GPCC and CHIRPS for their longer period. Although a thorough validation and comparison of these datasets across the AMS domain is missing, several studies have analysed one or more of these datasets in regions of the AMS (e.g. Franchito et al., 2009; Dinku et al., 2010; Trejo et al., 2016).

Lines 100-124: Note that the ocean model horizontal resolutions have not been listed.

The corresponding ocean model horizontal resolutions are now listed in these lines.

Line 129: Clarify whether this is surface temperature or surface-air (i.e. 1.5 or 2 metre)temperature that is being considered.

We are using 2-m near surface air temperature. Table 1 and this section now makes clarifies this point.

Line 132: The Welch t-test should be defined in the methods or referenced here. How does this differ from a student's t-test? Plus, how are the different ensemble members dealt with relative to this?

A reference and explanation about the use of the Welch t-test is now given in this line. The p-value is estimated from each ensemble member and these p-values are then combined into a single p-value using Fisher's method (Fisher, 1992). We have thus included the following the lines suggested by the reviewer:

"Only statistically significant differences are shown, according to a Welch t-test (Wilks, 2011), which accounts for the difference in sample size and variance between model and observations/reanalysis data. The significance for simulations with multiple ensemble members is estimated first for each ensemble member and then combined into a single probability or p-value using Fisher's method (Fisher, 1992)."

Line 137/138: Does the stronger Bolivian LLJ support a stronger seasonal cycle/monsoon in the region north of the equator in South America during boreal summer? (I.e. in the South America component of the NAMS.)

Although the regions north of the equator do experience a monsoonal climate as defined by Wang et al. (2017) they are not comprised in North American Monsoon Literature. The stronger Bolivian LLJ is suggestive of stronger moisture transport to the south-eastern part of the South American Monsoon and would also suggest more rainfall than observed in this region while also drying the Amazon. The manuscript now reads:

"The South America Low-Level Jet, the low-level northwesterly flow in Bolivia, observed in Figure 1a, is stronger in the simulations. This stronger than observed jet is suggestive of a stronger moisture transport to the La Plata Basin, with has been associated with a drying of the Amazon and positive precipitation anomalies at the exit region of the jet (Marengo et al., 2012; Jones and Carvalho, 2018)."

Line 147/148: The physical outcome of this needs to be made explicitly clear to the reader, namely it appears that the inclusion of Earth System processes makes no difference to the SAMS.C3

Done. The manuscript now reads in these lines:

"The inclusion of Earth System processes appears to make no improvement on the low-level circulation biases."

Line 149/150: A better summary of the changes in historical forcing (compared to the pre-industrial) needs to be described in lines 119-124 in order for the reader to be able to understand possible changes. Clearly the reader will know that global GHG emissions have increase, but what are the relevant/local patterns of aerosol emissions,land-use change etc. between the two experiments?

A more detailed description of the differences between the historical and piControl experiments is given

in section 2.2:

In contrast to the pre-industrial control experiments, the historical experiments use time-varying aerosol and greenhouse gas emissions and land-use change (Eyring et al., 2016). In Latin-America, land-use change for agricultural purposes have dramatically decreased tree cover in Central America and south-eastern Brazil since the 1950s (Lawrence et al., 2012), thereby affecting the surface energy balance. The regional emissions of carbonaceous aerosols, nitrogen oxides and volatile organic compound in Latin American megacities are also considered in the historical experiments. These emissions are noteworthy, e.g., due to the impact of black carbon emissions by increased biomass burning in the Amazon and northern Central America (Chuvieco et al., 2008).

Line 150-152: Given the length of the pre-industrial control integrations that are available (and given the small size of the forcing when compared to the historical experiment), the internal variability of quantities such as those listed here (and elsewhere through the results) within the pre-industrial should be considered as a means to understanding the significance of any change.

Both in the case of Figure 2e,f and Figure 7,8 h, the differences between historical and piControl have been shown only where the historical experiments shows a statistically significant difference from the piControl variability, as defined by a Welch t-test between the two experiments.

Line 175: I understand the logic, but the chosen model comparison mixing UKESM with the GC3 model appears rather unclean.

The choice of model comparison was based on the fact that the four low resolution simulations (GC3 N96-pi, GC3-hist, UKESM1-pi and UKESM-hist) were virtually indistinguishable for precipitation, ITCZ and low-level circulation biases. Comparing the low resolution coupled model with the medium resolution coupled or the AMIP simulation provides a better. For brevity, we now compare one coupled low resolution simulation with the coupled medium-resolution and the AMIP simulation. The manuscript now clarifies that this choice of simulations shown was based on showing the main biases and the differences.

Line 193: In what way is the low-level wind structure biased?

The manuscript now describes the wind biases:

The modelled low-level wind in the coupled model structure shows significant biases near the ITCZ. These wind biases are observed as stronger wind vectors converging toward the ITCZ during boreal summer and spring and stronger wind vectors diverging away from the equator during boreal winter.

Lines 171-222 and onwards: All of the comparisons whether maps or seasonal cycles would benefit from a table of quantitative comparisons between the various datasets, such as pattern correlations (or just correlations for the seasonal cycles) and RMSE.This is standard practice in multi-model evaluation studies.

Figures 1, 2, 7, 8 and 9 now show correlation and pattern correlation coefficients and RMSE.

Line 239: That the AMIP models "removed the spatial patterns" is strange wording. Did any bias remain at all? Generally, I think that this study could be significantly strengthened if a fuller comparison

could be made between AMIP runs of these two models (which will be available as contributions to the CMIP6 DECK) could be thor-ughly compared with the coupled historical runs. The absence of SST bias would make for improved understanding.

The manuscript now includes results from the HadGEM3 GC3.1 AMIP simulation run at low resolution N96. The discussion is now updated to highlight the biases that are removed when the SST biases are removed, mainly the dry Amazon bias. However, biases in Central American rainfall and in the North American Monsoon are not improved even with "the right" SSTs.

Line 253: Here the run is referred to as "high-resolution" yet in the abstract it was medium resolution. The consistency within the manuscript needs to be improved.Could the manuscript not also examine a higher resolution version of the GC3 experiment, e.g. at N512?

The manuscript is now consistent with the wording of Williams et al. (2018), Menary et al. (2018) as to refer to the simulations as low (N96) and medium (N216) resolutions. Indeed the higher resolution simulation could be examined and compared, although long runs, pre-industrial or historical, at that resolution are not available and can therefore not be directly compared to the experiments used in this study or with observations.

Line 262: Are there any published onset measures for the AMS that could be used tomeasure this? And how is the onset objectively defined from Figure 9b? E.g. 1mm/dthreshold, or the maximum rate etc.?

There are some studies that analyse onset and retreat timings in precipitation time series or using other metrics such as OLR in the AMS. However, most of the methods are not suited to address model output as the required fields are not all provided by the modelling centres or are tailored to be used in one specific dataset and using these methods in model output requires further statistical treatment. Ongoing research by the authors aims to cover this shortcoming in the literature by presenting a robust method that can use precipitation time series from different datasets (observations, model, reanalysis) and in different monsoon regions. To address the specific comment of the reviewer, the statement in the manuscript now clarifies that onset timing is merely qualitatively well represented by the models.

Lines 256-287: In the tropics, and especially for monsoons, I would expect the sea-sonal cycle of precipitation to be discussed in the context of the lower tropospheric circulation. This doesn't necessarily need to be done in the same paragraphs (the lay-out here is fine), but at the very least I would expect the discussions of precipitation biases here to reference the circulation biases for consistency. This is because of the intimate connection between circulation and precipitation in the tropics: winds providing moisture to the monsoon and the monsoon heating feeding back on the circulationto bring more moisture. At present the discussion is kept very separate. This could be aided by adding wind vectors to Figures 7 and 8.

The wind vectors have now been added to these Figures. Furthermore, the description of these figures and the discussions now couples the circulation and the precipitation to highlight, for instance, the relation-ship between the moisture transport away from the Amazon into southeastern Brazil and a corresponding

dry Amazon bias and a wet southeastern Brazil bias in these models.

Lines 256-287: It would be preferable to have some contextual comparison with other contemporary models (or at least CMIP5). How did CMIP5 perform for the NAMS and SAMS (cite references)? Do the UKMO models here fit within that envelope or arethey better/worse? This will help improve the level of interest in this study outside thesingle modelling group. Furthermore, can the authors state how the current UKMOmodel versions (especially GC3.1) have advanced upon earlier versions (HadGEM3,HadGEM2-ES, even HadCM3) with respect to the AMS? Are there any published works mentioning those models? It would be useful for the community to understand if the simulation is being improved or whether significant biases are persisting.

We now provide context to assess whether these models have improved, which biases have been removed and which have persisted, with references, in the discussion section and for each monsoon region. For example for the North American Monsoon:

These results suggest model improvement on the simulation of the North American Monsoon sfrom previous versions of the MOHC models (Arritt et al., 2000), and most of the model cohorts of CMIP3 and CMIP5 (Geil et al., 2013). For example, most of CMIP5 models showed a very wet bias during monsoon maturity whereas rainfall during monsoon maturity in all the experiments of this study within 1 mm day-1 of observations. However, these models continue to show biases during monsoon retreat as rainfall does not decrease as sharply as in observations after mid-September.

Line 288: In the deep tropics, OLR is not really going to tell us much more than wealready learn from precipitation, since much of the convection is deep. What is the nature of convection in the regions discussed? If any particular regions are dominated by shallow convection/warm rain, then this could be highlighted by references to relevant published works.

While we generally agree with this reviewer's comment that in the deep tropics during the wet season of each monsoon, OLR is highly correlated with precipitation and virtually indistinguishable, Fig. 10 does show interesting differences in OLR and $\omega$ between model and reanalysis that do not agree with the precipitation. Particularly in the MSD region, the first peak, MSD and second peak characteristics in precipitation do not agree with OLR. The OLR would suggest a relatively similar first peak magnitude in the simulations and a weaker second peak than observed, however, the simulated 'precipitation shows a significantly wetter (Fig. 9) first peak and very similar second peak compared to both TRMM and ERA5 precipitation. The analysis of OLR, $q$ and $\omega$ may point to model biases in the treatment of convection and potential feedbacks. The height of convection influences the radiative balance, whereas characterising the strength of ascending and mid-level moisture aids to evaluate several aspects of the model's convection and microphysical schemes.

Line 297: How certain can we be about the tropospheric moisture in any case in areanalysis? What level of data is assimilated in some of these remote regions? Can any ground-truthing (really air-truthing!) be performed (even if not shown) using nearbyRS launches such as those publicly available from Wyoming?

According to the Wyoming website and the NOAA station archive, regular soundings have been made in Manaus and Leticia (in the Amazon region), at Empalme, Sonora, México (core North American Monsoon), at Guatemala City and San Cristobal de las Casas, Mexico (in the MSD region) and at Sao Paulo and Brazilia (southeastern Brazil) at least since 1979. These radiosonde observations are assimilated twice a day into ERA-5. Although scarce and not as widespread as in other regions these are valuable input into ERA-5 and while not ground truthing over the whole domain of each monsoon region, these are the best estimates of tropospheric moisture available and therefore, arguably, makes ERA-5, and other reanalyses, a good standard to compare against. Analysing reanalysed and modelled tropospheric moisture where no RS launches are assimilated into the reanalysis would in fact be subject mostly to the reanalysis model driving the variables, for example over the ocean. We hope that showing that there have been RS launches assimilated into ERA5 in all the regions analysed of this study would answer this reviewer's concern. A more thorough comparison between ERA-5 and RS would be most beneficial but outside the scope of our study.

Line 328: Unlike the implication in the abstract, there is no assessment made hereof general ENSO behaviour in these coupled models – and if the driving point of ateleconnection is faulty then resultant impacts over the AMS will hardly do well. Asummary of the behaviour of ENSO in these models with reference to a publishedassessment of their performance should be made.

This is an interesting and recurrent point by this reviewer. Menary et al. (2018) showed that the EN3.4 index has a similar power spectrum in the pre-industrial control experiments (see Figure 1 of this document) when compared to observations. The models also show a good representation of the perturbation to the Walker circulation by ENSO events (see Figure 2 of this document). The main patterns of variability (Figure 3) are also reasonably reproduced, particularly by the medium-resolution simulation. Of particular importance to the study at hand is ENSO diversity and the impact each different ENSO event has on the rainfall of the American Monsoon System. The characteristics of ENSO in these models are now summarized in section 5 in the manuscript and several of these figures have been added to a Supplementary information document that accompanies the manuscript.

Line 332: Is this in units of temperature (degC/K) or a normalised index in terms ofstandard deviations? Where is the index taken from or how have you calculated it?

The index has units of K, and was estimated from the HadSST dataset in the El Niño 3.4 region, as described in line 333 of the original manuscript. The figure now shows the units of the index.

Line 334: What are the years included in the observed composites of El Nino and LaNina? Has the impact of CP and EP El Nino events been considered and what doesthe published literature say about the different impacts of such events on the NAMSand SAMS?

Cai et al. (2020) and references therein show that ENSO teleconnections to SAMS depend on the type of ENSO, as shown by their indices (see another response and figures 3-6 below). For instance, Figure 4 of this document shows that GC3 N216 has ENSO events well represented in all the quadrants of the PC

space. We now included a new figure (Figure 6 of this document and new Figure 13 of the manuscript) comparing observed and modelled responses to CP and EP ENSO events. This analysis would of course be improved by analysing ENSO diversity in future projections and providing a more thorough analysis of ENSO diversity, for instance, measuring the skewness in the PC space, but we considered this to be outside the scope of this study.

Line 351: It would be very instructive if wind vectors were added to Figure 12, enabling to reader to understand something of the mechanism by which ENSO controls rainfall anomalies in the AMS. The authors should then elaborate upon this in the text.

We agree with this reviewer's comment, we added the wind vectors to this Figure, but found that this would overcrowd the figure and decrease clarity. It is important to note that the mechanisms for the teleconnection to the subtropics are different from the teleconnection to the equatorial Amazon and therefore the most relevant wind anomalies for each teleconnection take place at different levels of the atmosphere. Figure 2 of this reply shows the Walker circulation anomalies in circulation and moisture. The Amazon region anomalies are closely related to this perturbation to the overturning circulation whereas the subtropical regions are affected by the perturbation of ENSO to Rossby wave-trains and the subtropical jets. We have added this figure to the supplementary material.

Line 348-350: It's not immediately obvious how the NAO links described are relevant to the study at hand. The authors should either make this clear or remove this text.

The boreal winter-time NAO has been shown to influence precipitation in Central American and the Caribbean (Giannini et al., 2000), as well as northern South America (Giannini et al., 2004). Therefore, capturing the response in the North Atlantic SLP field may be important to capture secondary aspects of ENSO teleconnections. The link is now explained in the manuscript as:

"While the models seem to be able to capture this response of the NAO, the simulated response is weaker than observed, which may be relevant to simulate a secondary effect of the NAO on Central American and northern South American rainfall (Giannini et al., 2000, 2004)."

Lines 365-370: The authors should consider whether the lack of nonlinearity in the modelled ENSO response reflects the lack of diversity of simulated ENSO in the model(e.g. the lack of distinct central Pacific or east Pacific events).

Figures 5 and 4 of this document show the spread of boreal winter SST patterns as measured by the principal component analysis, as shown by (Cai et al., 2020). The variability of these models appears to cover a range of ENSO diversity, except perhaps missing extreme events such as the 97-98 and 2015-2016. The patterns associated with Central and East Pacific positive ENSO events (Figure 5) agree with the patterns of HadSST. These figures have now been added to the Supplementary material and the main results are introduced in section 5 of the new manuscript, to validate the fact that ENSO diversity seems, to a first degree, well represented in these models. A more thorough analysis would better evaluate how different are EP and CP in the observations and in the simulations, using metrics such as skewness or

perhaps the degree of coupling of the SSTs to the Walker circulation.

In any case, the teleconnections of the different types of ENSO events to South America (shown in Figure 6) appear to be independent of the ENSO type in the simulations. This may be because the model diversity is not representative of the observed ENSO diversity in other metrics, or perhaps because the simulated SST patterns do not couple to the atmospheric circulation in the same way, but this warrants further analysis, outside of the scope of this study.

Line 376: The authors could be more explicit on the likely kink between cloud cover and the warm bias in the SAMS domain. If precipitation is too weak, this should be stated explicitly. (Note there would also be a soil-moisture feedback as a result.)

The manuscript now makes the link between precipitation, cloud cover and temperature explicit:

In the Amazon, the simulations showed a warm bias (+2 K) during austral spring and summer, a typical feature of previous models (Jones and Carvalho, 2013), and a colder than observed southeastern Brazil. These biases were linked with decreased cloud cover and less rainfall over the Amazon and more high clouds and rainfall in southeastern Brazil (Figures 7 and 9). The low cloud cover, warm and dry Amazon biases are intertwined with the low-level circulation from the Atlantic into the South American continent. The biases in the circulation during austral summer were observed as a northerly flow anomaly over the central and southern Amazon, a feature that has been associated with a stronger moisture transport away from the Amazon (Marengo et al., 2012; Jones and Carvalho, 2018). During the period of maximum mean rainfall rates in February, the simulations can overestimate rainfall by 3 mm day -1 in southeastern Brazil and underestimate rainfall in the Amazon by a similar rate.

Lines 376-380: Finish the sentence by making explicit how the land-sea temperaturecontrast may feedback on the monsoon.

Addressed in the previous comment.

Line 391: Make explicit whether the Ryu and Hayhoe study was using CMIP5 models.

Yes, the study used CMIP3 and CMIP5 models, the manuscript now makes this explicit.

Line 393: With reference to the earlier comment on the abstract, the authors shouldavoid the terminology of intraseasonal variability here since the MJO/BSISO have not been assessed. Done.

Lines 371-404: In the conclusions I would want to see a more thorough synthesis ofthe results (e.g. how all the meteorological components fit together) than a summaryof each in turn. It would also be worth reflecting upon (if possible) how these modelssit in comparison to published literature on the AMS in CMIP3/5 models or on earlier versions of UKMO models.

The discussion section has been changed significantly to address this comment. The new manuscript now discusses each region of the AMS separately; for each region a summary of the biases in circulation, temperature and precipitation is given, indicating where they might be linked and finally whether these CMIP6 versions are an improvement from previous versions of the UKMO and CMIP5 models.

Line 413: See earlier comments on higher/medium resolution. Done.

Line 418: Need to see a summary of how the Earth System processes influence the response to forcing.

While we do not provide a thorough summary of the Earth System processes, as we did not investigate them explicitly and may be outside the scope of our study, the manuscript does state:

"A relevant difference between UKESM1 and GC3 is that warming over the historical period in Mexico and the Amazon is higher in UKESM1 than in GC3. This warming may be a consequence of the land-use change in these regions playing a role in the UKESM1 representation of soil-atmosphere feedbacks."

Figure 1: The domains used later in Figure 3 etc. need to be pictured somewhere, e.g.on this figure.

The domains are now shown in Figures 1, 7 and 8.

Trivia:

Lines 13/14: Perhaps replace "in subtropical America" [meaning USA?] with "inthe subtropical Americas".

To avoid confusion with native english speakers, we have opted to use the term "subtropical North and South America".

Line 21: Change "copuled" to "coupled". Done.

Line 42:"...and the dynamics the features largely characterise the MSD characteristics...". I don't understand what is meant here, something is wrong with the grammar.

Sentence has been reworded.

Line 43: Change "reproduce accurately" to "accurately reproduce".

Line 51: Remove hyphen from "South-America". Done.

Line 66: Space needed in "Met Office". Done.

Line 119: Replace "beginning for" with "covering"; replace "that include" with "of". Done.

Line 142: Change "temperature" to "temperatures". Done.

Line 171: Second "the" is not required. Done.

Line 184: brackets not needed around location point. Done.

Line 195: By "a minimum" do you mean "southernmost position"? This would be easier to understand. Changed for "southernmost position".

Line 302: Replace "indicative" with "are indicative". Done.

Line 304: Clarify if the decreased omega is a reduction or increase in ascent.

Line 309: Mixture of singular and plural in this line. Done.

Line 331: By convention, "El" is not included when referring to the "Nino-3.4 index". Done.

Line 362: Change "opposite sign response" to "opposite signed response". Done.

**References**

Adams, D. K. and Comrie, A. C. (1997), 'The north American monsoon', *Bulletin of the American Meteorological Society* **78**(10), 2197–2214.

Arritt, R. W., Goering, D. C. and Anderson, C. J. (2000), 'The north american monsoon system in the hadley centre coupled ocean-atmosphere gcm', *Geophysical Research Letters* **27**(4), 565–568.

Cai, W., McPhaden, M. J., Grimm, A. M., Rodrigues, R. R., Taschetto, A. S., Garreaud, R. D., Dewitte, B., Poveda, G., Ham, Y.-G., Santoso, A. et al. (2020), 'Climate impacts of the el niño–southern oscillation on south america', *Nature Reviews Earth & Environment* **1**(4), 215–231.

Fisher, R. A. (1992), Statistical methods for research workers, *in* 'Breakthroughs in statistics', Springer, pp. 66–70.

Geil, K. L., Serra, Y. L. and Zeng, X. (2013), 'Assessment of cmip5 model simulations of the north american monsoon system', *Journal of Climate* **26**(22), 8787–8801.

Giannini, A., Kushnir, Y. and Cane, M. A. (2000), 'Interannual variability of Caribbean rainfall, ENSO, and the Atlantic Ocean', *Journal of Climate* **13**, 297–311.

Giannini, A., Saravanan, R. and Chang, P. (2004), 'The preconditioning role of tropical Atlantic variability in the development of the ENSO teleconnection: Implications for the prediction of Nordeste rainfall', *Climate Dynamics* **22**, 839–855.

Jones, C. and Carvalho, L. M. (2013), 'Climate change in the South American monsoon system: present climate and CMIP5 projections', *Journal of Climate* **26**(17), 6660–6678.

Menary, M. B., Kuhlbrodt, T., Ridley, J., Andrews, M. B., Dimdore-Miles, O. B., Deshayes, J., Eade, R., Gray, L., Ineson, S., Mignot, J., Roberts, C. D., Robson, J., Wood, R. A. and Xavier, P. (2018), 'Preindustrial control simulations with hadgem3-gc3. 1 for cmip6', *Journal of Advances in Modeling Earth Systems* **10**(12), 3049–3075.

Vera, C., Higgins, W., Amador, J., Ambrizzi, T., Garreaud, R., Gochis, D., Gutzler, D., Lettenmaier, D., Marengo, J., Mechoso, C. R., Nogues-Paegle, J., Dias, P. L. S. and Zhang, C. (2006), 'Toward a unified view of the American monsoon systems', *Journal of Climate* **19**(20), 4977–5000.

Wang, P. X., Wang, B., Cheng, H., Fasullo, J., Guo, Z., Kiefer, T. and Liu, Z. (2017), 'The global monsoon across time scales: Mechanisms and outstanding issues', *Earth-Science Reviews* **174**, 84–121.

Williams, K. D., Copsey, D., Blockley, E. W., Bodas-Salcedo, A., Calvert, D., Comer, R., Davis, P., Graham, T., Hewitt, H. T., Hill, R., Hyder, P., Ineson, S., Johns, T. C., Keen, A. B., Lee, R. W., Megann, A., Milton, S. F., Rae, J. G. L., Roberts, M. J., Scaife, A. A., Schiemann, R., Storkey, D., Thorpe, L., Watterson, I. G., Walters, D. N., West, A., Wood, R. A., Woollings, T. and Xavier, P. K. (2018), 'The met office global coupled model 3.0 and 3.1 (gc3. 0 and gc3. 1) configurations', *Journal of Advances in Modeling Earth Systems* **10**(2), 357–380.

[Figure]

Figure 1: Power spectrum of the ENSO 3.4 index in pre-industrial control simulations of the HadGEM3 and UKESM1 models and HadSST data. The gray lines indicate the 2 and 7 yr period.

[Figure]

Figure 2: DJF Longitude-height Walker circulation anomalies of specific humidity (colour-contours), $\omega$ (vectors) and zonal wind (line-contours) during El Niño events (left) and La Niña events (right). Results are shown for ERA-5 (upper), UKESM-pi (middle) and HadGEM3 piControl (lower).

[Figure]

Figure 3: SST patterns [arbitrary units] of the two leading EOFs in HadSST, GC3 N216 and UKESM1.

[Figure]

Figure 4: Principal component (PC) space of the first and second leading PCs of the deseasonalized Pacific SSTs diagram showing HadSST (circles) and GC3 N216 (triangles). The PCs are showing as DJF-means.

[Figure]

Figure 5: SST anomalies [K] for East Pacific (EP) and Central Pacific El Niño events in HadSST, GC3 N216 and UKESM piControl. EP (CP) events were defined where the E-index (C-index) was greater than 1. In the bottom panel, the frequency of events per decade (with standard deviation as error bar) is shown for HadSST and the simulations used in this study. The E-index is computed from $(PC1 - PC2)/\sqrt{2}$ and the C-index from $(PC1 + PC2)/\sqrt{(2)}$.

[Figure]

Figure 6: Precipitation anomalies in GPCC 1940-2013, GC3 N216, GC3 N96-pi and GC3 AMIP for the four different types of ENSO events, as defined by Cai et al. (2020). Statistically significant anomalies (95% confidence level) are hatched.

---

## Referee Report (RR1)

Review of *The American Monsoon System in HadGEM3.0 and UKESM1CMIP6 simulations*
 by Garcia-Franco et a

David K. Adams
dave.k.adams@gmail.com

Recommendation:  Minor Revision

**General Comments**

The authors have adequately addressed my concerns with respect to amplifying a bit the general context for these type of model run comparison studies.   At this stage, I feel the paper is essentially ready for publication.  I have a list of small changes that should be made as well as a new suggested reference which would be good for the discussion on  Central Amazon precipitation.

**Minor Comments**

You say "well represented" numerous times, perhaps you can change the description a bit.

Line 7  I like "notable" better than "noticeable"

Line 13  The precipitation biases over the Amazon and southeastern Brazil are absent…
"Removed" sounds like you actively took them out.

Line 20  Rewrite this sentence.  It is unclear.

Line 42.  I think another good reference for central Amazon rainfall that I forgot to mention in my previous review is Tanaka et al.  2014

Line 44   write "...greatly influence the South American ..."

Line 45   write "...demonstrated by observations…

Line 56  write … as in previous studies...
Line 60  Another particular relevant article with respect to modeling of the American Monsoons is our recent review  (Pascale et al. 2019).

Line 78  Check spelling "characteristics"

Line 109  write "...given their longer period."

Line 113  Define  "MOHC"

Line 119  What are "CMIP6 deck experiments"?   Baseline experiments/runs?

Line 163 Write "...which has been associated..."

Line 164  Write  "In turn, …"

Line 208  This sounds strange to my ears  "The EP ITCZ reaches minimum precipitation"

Line 210  Write "The low-level winds are predominantly easterly"
Easterlies is a proper noun typically refering to those climatological winds.

Line 236  Correct "northwest-southeast"

Line 266  Probably better to write "eastern Brazil"

Line 268  Write "are different from the observations."

Line 443 Maybe write "...has greater climate sensitivity..."

Line 448  Correct "entrainment"

**References**

Pascale, S., Carvalho, L.M.V., Adams, D.K. *et al.* Current and Future Variations of the Monsoons of the Americas in a Warming Climate. *Curr Clim Change Rep* 5, 125–144 (2019). https://doi.org/10.1007/s40641-019-00135-w

Tanaka, L.M.d.S., Satyamurty, P. and Machado, L.A.T. (2014), Diurnal variation of precipitation in central A mazon B asin. Int. J. Climatol., 34: 3574-3584. doi:10.1002/joc.3929